



# Condensation/immersion mode ice nucleating particles in a boreal environment

Mikhail Paramonov[1,a], Saskia Drossaart van Dusseldorp[1], Ellen Gute[2], Jonathan P. D. Abbatt[2], Paavo Heikkilä[3], Jorma Keskinen[3], Xuemeng Chen[4,5], Krista Luoma[4], Liine Heikkinen[4], Liqing Hao[6], Tuukka Petäjä[4] and Zamin A. Kanji[1]

[1]Institute for Atmospheric and Climate Science, ETH Zürich, Switzerland
[2]Department of Chemistry, University of Toronto, Toronto, Ontario, Canada
[3]Aerosol Physics laboratory, Physics unit, Tampere University, Tampere, Finland
[4]Institute for Atmospheric and Earth System Research/Physics, Faculty of Science, University of Helsinki, Helsinki, Finland
[5]Institute of Physics, University of Tartu, Tartu, Estonia
[6]Department of Applied Physics, University of Eastern Finland, Kuopio, Finland

[a]now at: Finnish Meteorological Institute, Helsinki, Finland

*Correspondence to*: Mikhail Paramonov (mikhail.paramonov@fmi.fi) and Zamin A. Kanji (zamin.kanji@env.ethz.ch)

**Abstract.** Ice nucleating particle (INP) measurements were performed in the boreal environment of southern Finland at the Station for Measuring Ecosystem-Atmosphere Relations SMEAR II in the winter-spring of 2018. Measurements with the Portable Ice Nucleation Chamber (PINC) were conducted at 242 K and 105% relative humidity with respect to water. The median INP number concentration [INP] during a six-week measurement period was found to be 13 L$^{-1}$. [INP] spanned 3 orders of magnitude and showed a general increase from mid-February until early April. No persistent local or regional sources of INPs in the boreal environment of southern Finland could be identified. Rather, it is hypothesised that the INPs at SMEAR II are a result of dilution during long-range transport. Despite high variability, the measured [INP] values fall within the range expected for INP number concentrations measured elsewhere at similar thermodynamic conditions. [INP] did not correlate with any of the examined relevant parameters during the entire field campaign, indicating that no one single parameter can be used to predict the INP number concentration at the measurement location during the examined time period. The absence of correlation across the entire field campaign also suggests that a variety of particles are acting as INPs at different times, although it was indirectly determined that, on average, ambient INPs are most likely in the size range of 0.1–0.5 μm in diameter. On shorter time scales, several particle species correlated well with [INP] implying their potential role as INPs. Depending on the meteorological conditions, signatures of black carbon (BC), supermicron biological particles and sub-0.1 μm particles, most likely nanoscale biological fragments such as ice nucleating macromolecules (INMs), have been found in the INP signal. However, an increase in the concentration of any of these particle species may not necessarily lead to the increase in [INP], reasons for which remain unknown. Limitations of the instrumental setup and the necessity of the future field INP studies are addressed.



# 1 Introduction

Atmospheric aerosol particles play an important role in the global climate by influencing the Earth's hydrological cycle, energy
and radiation balance. Due to their importance, aerosol-cloud interactions have been a subject of intense research over the last

several decades (e.g. Twomey, 1974; Lohmann and Feichter, 2004; DeMott et al., 2010; Kerminen et al., 2012). Despite that,
exact quantification of aerosol effects on the changing cloud properties and the ability to predict future climate based on
expected changes to global aerosol burden have been challenging (Boucher et al., 2013).

Ubiquitous in the atmosphere, aerosol particles are responsible for the formation of liquid and ice clouds by acting as cloud
condensation nuclei (CCN) and ice nucleating particles (INPs), respectively. Both have received a lot of attention in recent

years, with a multitude of studies attempting to quantify the importance of aerosols in the aerosol-cloud-climate system. The
warm cloud regime, i.e. formation of liquid droplets on CCN, is understood fairly well (Andreae, 2009 and references therein;
Paramonov et al., 2015); however, establishing the exact connection between CCN and cloud droplet number concentration
has remained challenging (Moore et al., 2013). At the same time, the cold and mixed-phase cloud regimes, i.e. formation of
ice crystals on INPs, present many open questions (DeMott et al., 2011; Kanji et al., 2017). Ice crystals in the atmosphere can

form homogeneously, i.e. by freezing of pure water drops in the absence of any insoluble foreign substances. Such process
requires temperatures below -37°C (e.g. Murray et al., 2010). In the temperature range between -37°C and 0°C ice can form
heterogeneously, i.e. when the freezing is aided by an INP. Heterogeneous ice nucleation has four known mechanisms:
deposition nucleation, condensation freezing, immersion freezing and contact freezing (Vali et al., 1985; Vali et al., 2015);
although the significance and prevalence of each individual mechanism in the atmospheric ice nucleation are still under debate.

More recently it has also been shown that pore condensation and freezing could be alternative to deposition nucleation
(Marcolli, 2014; David et al., 2019).

The difficulty in understanding the ice nucleation (IN) processes in the atmosphere is associated with the rarity of ambient
INPs (DeMott et al., 2010), spatio-temporal variability of particle species known to be good INPs (Boose et al., 2016a; Welti
et al., 2018), elusiveness of exact particle properties leading to atmospheric ice nucleation (Knopf et al., 2014; Paramonov et

al., 2018) and secondary ice production mechanisms (Hallett and Mossop, 1974; Field et al., 2017). Atmospheric INP number
concentrations [INP], while increasing with decreasing temperature, reach maximum values of 1000–10000 L$^{-1}$ (DeMott et al.,
2010), which is orders of magnitude lower than CCN number concentrations or total aerosol particle number concentrations
(Paramonov et al., 2015). At warmer sub-zero temperatures, where the nucleation rate is low and at which first ice nucleation
events take place, [INP] values are extremely small, e.g. ~10$^{-6}$ L$^{-1}$ (Petters and Wright, 2015). Such low concentrations make

it difficult to identify these particles and assess their atmospheric relevance. Several particle species are well-known to act as
efficient INPs under certain atmospheric conditions, such as mineral dust (e.g. Cantrell and Heymsfield, 2005), biological
aerosols (e.g. Després et al., 2012 and references therein), black carbon (BC; e.g. DeMott, 1990) and marine organic aerosol
(e.g. Ladino et al., 2016; McCluskey et al., 2017). Even for these well-known INP species, their exact properties responsible
for ice nucleation, i.e. active sites, are not well-understood (Kanji et al., 2017). Previous studies have alluded to ambient INPs





not being of any particular chemical composition or predisposition to ice nucleation at all (Knopf et al., 2014; Paramonov et al., 2018). Additionally, even if [INP] is accurately determined, there is a disconnect, often on the orders of magnitude, between the INP number concentration and the ice crystal number concentration (ICNC) (Cantrell and Heymsfield, 2005). Several secondary ice production mechanisms are known to contribute to the elevated ICNC as compared to INP; however, exactly how the ICNC will respond to changes in INP, if at all, remains unresolved. Despite these uncertainties, it has been shown that

ice is present in significant amounts in various cloud types all around the globe (Lau and Wu, 2003; Sporre et al., 2014), and ice most certainly plays a non-trivial role in cloud radiative properties and their response to phase changes (e.g. Lohmann, 2002).

In order to probe the INP number concentration in various environments, a multitude of field measurements have taken place within the last decade utilising the latest technological developments, with an overview of these measurement endeavours

presented in Kanji et al. (2017). The studies have shown that across all sub-freezing temperatures ambient [INP] varies across 10 orders of magnitude, and, as expected from the Classical Nucleation Theory (CNT; aufm Kampe and Weickmann, 1951), a decrease in temperature leads to an increase in [INP]. Despite the wide range of ambient [INP] values and types, at any given temperature [INP] typically varies over ~4 orders of magnitude, reaching values as high as $> 10^3$ $L^{-1}$ at temperatures just above homogeneous freezing. This somewhat restricted range of ambient [INP] values coupled with the increasing knowledge of

INP properties has led to the development of parameterisations used to estimate ambient [INP] (Richardson et al., 2007; DeMott et al., 2010; Tobo et al., 2013). Field studies, therefore, serve as both basis and validation for these parameterisations. Among the environments that have been selected for previous INP field studies, boreal forest stands out as having insufficient data on INP properties and processes. Boreal forest covers 7–8% of the total continental area, and it spreads across North America, Asia and Northern Europe between 50°N and 70°N with local variation (Olson et al., 1983). Yet, the majority of

ambient INP field measurements have focused on areas with significant concentrations of known INP species, e.g. desert outflow regions (e.g. Boose et al., 2016a; Welti et al., 2018) and the Southern Ocean (McCluskey et al., 2018). Therefore, the questions of what type of particles act as INPs and how they affect cloud properties in the boreal environment remain open. In order to bridge the gap in the knowledge of IN processes and characteristics in the boreal forest, a field campaign took place in southern Finland in winter-spring 2018. The objectives of the campaign included the quantification of [INP] in

condensation/immersion freezing modes under mixed-phase cloud conditions, the comparison to previously published data from other locations around the globe, the investigation of INP physical and chemical properties and the identification of the already known INP species in the boreal environment.

## 2 Methodology

### 2.1 Measurement location

Ice nucleation measurements presented in this study took place in Hyytiälä, southern Finland, the location of the Station for Measuring Ecosystem-Atmosphere Relations SMEAR II (61° 50' 50.685"N, 24° 17' 41.206"E, 181 m a.m.s.l.). SMEAR II is a



comprehensive measurement station with a multitude of continuous online and offline measurements of various gas, aerosol, soil, meteorological and radiation parameters (Hari and Kulmala, 2005). The station is located in a boreal coniferous forest surrounded mostly by Scots pine. The nearest city of Tampere (pop. 220 000) is located 50 km southwest of the station, and,

therefore, the station is considered to be a rural background site. SMEAR II experiences both continental and maritime air masses, although particle number concentrations are typically low (Sogacheva et al., 2005).

The measurement campaign lasted from 19 February until 2 April 2018 and took place as part of the HyICE-2018 measurement activities performed by several national and international research groups.

**2.2 Instrumentation and setup**

The basic instrumental setup can be seen in Fig. 1. Ambient air was drawn through a vertical ~2 m tall inlet mounted outside of the building. The inlet was heated to 25–30°C and sampled ambient air with a flow rate of 250 Lpm. The inlet was heated in order to evaporate droplets and ice crystals entering the measurement setup, as well as to avoid potential condensation of water vapour on the inner surfaces of the tubing and instruments. The sample flow then entered the Portable Fine Particle Concentrator (PFPC), described by Gute et al. (2019) and based on the design by Sioutas et al. (1995). PFPC efficiently

concentrates aerosol particles up to a factor of 21±5 when operated at low altitude and in horizontal configuration, with a size-dependent enrichment factor where larger particles are concentrated more efficiently than the smaller ones (Gute et al., 2019). In this study the size-dependent concentration factor was determined for ambient particles of various sizes by measuring total particle number size distribution before and after the PFPC. The PFPC was used in order to improve the signal-to-noise ratio and to allow for longer duration of INP measurements. The output flow of the PFPC is ~10 Lpm. For non-concentrated

measurements, the PFPC was bypassed, and ambient air was sampled directly through the heated inlet. Since large particles entering the measurement setup can be mistaken for ice crystals, an impactor with a cut-off size of 2.5 μm was installed inside the PFPC, and a cyclone with a similar cut-off size was used for ambient measurements during the first half of the campaign. During the second half of the campaign the impactor was removed, and both concentrated and non-concentrated air passed through the same said cyclone. As seen in Fig. 1, the sample flow passed through a molecular sieve drier in order to reduce

the relative humidity. The flow was then split in four parts. A condensation particle counter (CPC, TSI model 3010) used 1 Lpm to determine the total number of particles entering the measurement setup. An aerodynamicl particle sizer (APS, TSI model 3321) used 1 Lpm to determine the size distribution of particles in the size range of 0.5–20 μm in aerodynamic diameter. A differential mobility analyser (DMA) connected to a CPC (TSI model 3772) used 1 Lpm to determine the size distribution of aerosol particles in the size range of 0.01–0.5 μm in electrical mobility diameter. Another 1 Lpm was used for the INP

measurements by the Portable Ice Nucleation Chamber (PINC) and an optical particle counter (OPC, Lighthouse 5104 Remote).

PINC is a continuous flow diffusion chamber (CFDC)-type instrument (Rogers, 1988) used for online INP measurements. It has been used for both laboratory (e.g., Kanji et al., 2013; Paramonov et al., 2018) and field studies (e.g., Boose et al., 2016a). The main chamber of PINC consists of two parallel walls coated with a thin layer of ice. The temperature of both walls can be



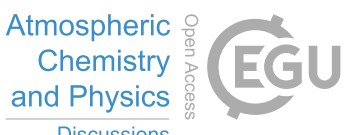

controlled in such a manner that a temperature gradient can be established between the walls while maintaining the desired temperature close to the centerline of the chamber. The ice on the walls and the applied temperature gradient result in a water vapour supersaturation between the walls, with a maximum close to the centerline of the chamber. When the temperature of both walls is the same, relative humidity with respect to ice ($RH_i$) is 100% and the chamber environment is subsaturated with respect to water. As the temperature gradient is applied, relative humidity with respect to water ($RH_w$) can increase to over

100%. Thus, PINC allows to perform an RH scan at a constant temperature in both sub- and supersaturated conditions with respect to water. The $RH_w$ of 100% also delineates what is considered to be the boundary between deposition nucleation ($RH_w$ < 100%) and condensation/immersion freezing regimes ($RH_w$ > 100%); although a more accurate differentiation between these regimes is not possible. The sample flow of 1 Lpm is guided by two particle-free sheath flows, 4.5 Lpm each. As aerosol particles travel close to the centerline of the chamber in a quasi-laminar flow, some particles activate as INPs, some as CCN

and some will remain unactivated. The total residence time (i.e. the time for activation and growth) of particles inside the chamber is nominally 7 seconds. In order to avoid miscounting liquid droplets for ice crystals, an evaporation section follows the main chamber. In this section the walls are held at the same temperature, and, therefore, $RH_i$ is 100% and $RH_w$ is below 100%. Under these conditions droplets evaporate and ice crystals are preserved. An OPC downstream of PINC measures the size distribution of all particles exiting the chamber. Based on the diffusional growth calculations (Rogers and Yau, 1989), a

size threshold of ~3.5 μm was determined, beyond which all particles are assumed to be ice crystals. The number of particles above this size is then considered the INP number concentration [INP]. The temperature uncertainty in the laminar flow of PINC is ±0.4 K, which is equivalent to the $RH_w$ uncertainty of ±2% (Chou et al., 2011).

        PINC is an online instrument and it measures [INP] in real time. However, the duration of its operation and the limit of detection (LOD) are limited by the quality of ice on the walls, which deteriorates over time, especially at high $RH_w$. Large ice

crystals and frost particles can fall off the walls and be erroneously miscounted as activated INPs. The concentration of these internally produced ice crystals is commonly referred to as the background signal, and it is measured by redirecting the main sample flow through a HEPA filter. The LOD of PINC is calculated as the error of the background concentration. At some point, normally after four to five hours of measurements, the background signal becomes higher than the sample signal. At this point the ice needs to be melted, chamber purged and walls re-iced. One icing cycle, which typically lasts four to five

hours, is hereafter referred to as an experiment. One or two experiments per day were conducted throughout the campaign, always between 10:00:00 and 23:00:00 local time (UTC+2).

        For the duration of the entire measurement campaign PINC was measuring [INP] at a temperature of -31°C (242 K) and at an $RH_w$ of 105%, i.e. in condensation/immersion freezing mode. The chosen measurement conditions are to simulate mixed-phase cloud conditions and to ensure that all IN-active particles are exposed to droplet activation conditions (DeMott et al., 2015).

The measurement conditions also allowed for a significant fraction of particles activating into ice crystals and the ability to compare the measured data to previously published results (Boose et al., 2016a; Boose et al., 2016b).

        Starting on March 2[nd], the aerosol chemical composition was measured with a long time-of-flight aerosol mass spectrometer (L-ToF-AMS) which is built upon the same characteristics as the high resolution ToF-AMS described in DeCarlo et al. (2006).



However, the time-of flight chamber in the L-ToF-AMS is longer, enabling the resolution to approach 8000 M/ΔM, which
helps with separating and identifying different peaks located close to each other in the mass spectrum. Due to the small
sampling flow rate of the instrument, an overflow of 3 Lpm was used to avoid losses in the inlet line. The sampling line also
included a Nafion dryer with 4 Lpm dry air flushing in order to keep the RH below 30%. The L-ToF-AMS was operating ~100
m away from the main INP measurement location. The low size for 100% transmission efficiency of the L-ToF-AMS is about
45 nm in electrical mobility (Liu et al., 2007). A $PM_{2.5}$ cyclone was mounted upstream of the inlet line to avoid clogging of
the instrument critical orifice (diam. 100 μm). The L-ToF-AMS ionisation efficiency (IE) was calibrated using atomised, dried
and size-selected 300 nm ammonium nitrate particles. A CPC (TSI 3772) was used as the reference instrument. The IE
calibration was performed at the beginning of the AMS measurements. The AMS data were analysed using standard ToF-
AMS data analysis toolkit (Squirrel V1.61B and PIKA1.21B). For mass concentration calculation, the default relative
ionisation efficiency (RIE) values 1.1, 1.2, 1.3 and 1.4 for nitrate, sulfate, chloride and organic, respectively, were applied.
The RIE for ammonium was 3.7, determined from IE calibration. A composition-dependent collection efficiency (CE) was
applied based on the principle proposed by Middlebrook et al. (2012).

Starting on March 11$^{th}$, the wideband integrated bioaerosol sensor (WIBS-NEO; DMT) was installed directly downstream of
the heated inlet in order to measure the concentration of biological fluorescent particles above 0.5 μm in diameter in the
ambient air. WIBS excites the particles with a laser and records the emission from fluorescent particles in the wavelength of
0.31–0.65 μm (Toprak and Schnaiter, 2013).

Besides the instruments described above, which were deployed specifically during the field campaign, several other datasets
have been used to complement the ice nucleation measurements. Most of these data are measured continuously at the SMEAR
II station and are freely available via the open research data portal AVAA (https://avaa.tdata.fi/web/avaa/etusivu; Junninen et
al., 2009). The auxiliary datasets used include meteorological data from SMEAR II (temperature $T$, pressure $P$, relative
humidity RH, wind speed WS, wind direction and cloud base height), aerosol particle size distributions in the size range of
0.002–20 μm measured by the local differential mobility particle sizer (DMPS) and APS, total aerosol particle number
concentration measured by the local CPC (Ntot), and black carbon BC concentration measured by the multi angle absorption
photometer (MAAP). It should be noted, however, that most of these data were not measured directly at the same location as
the INP measurements; rather, in the ~50 m vicinity.

## 2.3 Data treatment

As mentioned in the previous section, the measurement signal in PINC can be significantly affected by the background signal,
i.e. ice crystals and frost particles produced internally by the iced walls. To account for the background signal, background
measurements are taken before and after each sample measurement. The two background signal data points are linearly
interpolated across the ambient sample measurement time and subtracted from the corresponding ambient signal. PINC records
a data point every 12 seconds, and a single background-corrected [INP] data point can fall into one of the three categories:

1.    [INP] can be above LOD,





2.    [INP] can be positive and below LOD,

3.    [INP] can be negative.

Normally, only those [INP] values that are above LOD are reported, i.e. $[INP]_{excl<LOD}$. However, since atmospheric INP number

concentrations are typically low (DeMott et al., 2010), such way of reporting [INP] is biased towards high values and would

omit potentially significant low INP number concentrations. In order to better represent ambient [INP] and include [INP]

values below LOD, an approach suggested by Boose et al. (2016b) and Lacher et al. (2017) was adopted. In this approach, the

[INP] values falling in category 2 (0 < [INP] < LOD) are given the benefit of the doubt and are assumed to hold. [INP] values

falling in category 3 ([INP] < 0) are replaced with the minimum possible detectable value during the sampling time. This value

was calculated by dividing one INP ([INP] = 1) by the total volume of air sampled during the sample period. INP concentrations

calculated using this approached are denoted as $[INP]_{incl<LOD}$. In order to account for the high variability of the background

and sample signals, INP number concentrations are typically reported as 20-minute averages. A background measurement

before and after each 20-minute sample measurement is taken for 10 minutes. During the last several days of the campaign the

time of each sample and background measurement was reduced to 15 and 7.5 minutes, respectively.

From February 24$^{th}$ until March 21$^{st}$, the PFPC was used in order to increase the signal-to-noise ratio and to allow for longer

INP measurements. At the beginning of each experiment a concentrated measurement point $[INP]_{conc}$ was taken first. This was

followed by an ambient measurement point $[INP]_{amb}$ when PFPC was bypassed. These first two measurement points were then

used to calculate an enrichment factor EF for each experiment:

$$EF = \frac{[INP]_{conc}}{[INP]_{amb}}. \tag{1}$$

$[INP]_{conc}$ was measured for the remainder of each experiment, and the EF was then used to back-calculate $[INP]_{amb}$. The use

of EF assumes that no major changes of air masses and aerosol size distributions took place during each experiment when the

EF was measured and applied to the $[INP]_{conc}$ data. Towards the end of the campaign an improved icing procedure and a fairly

high $[INP]_{amb}$ signal allowed for measurements without the concentrator. During the last several days of the campaign only

$[INP]_{amb}$ was measured.

It has been reported in published literature that deviations of particles from the laminar flow are possible in the ice nucleation

chambers of the CFDC-type (DeMott et al., 2015; Garimella et al., 2017). In such cases a fraction of aerosol particles outside

of the lamina experiences $RH_w$ conditions below those intended. This leads to the particles outside of the laminar flow not

activating into ice crystals or water droplets and a consequent underestimation of the [INP]. Correction factors of 3 (DeMott

et al., 2015) and 1.4−9.5 (Garimella et al., 2017) have been suggested in order to account for the particle deviations outside of

the laminar flow. Correction factor tests with PINC have been performed in the laboratory setting prior to the field

measurements in order to determine the correction factor that is specific to PINC and to the field measurement conditions (T

= -31°C and $RH_w$ = 105%). The correction factor was determined to be 1.142, and all measured and reported INP





concentrations have been multiplied by this correction factor. The details of the correction factor tests can be found in Drossaart van Dusseldorp (2018).

## 3 Results and discussion

INP measurements with PINC took place between 21 February and 1 April 2018. Weather conditions throughout the campaign can be generally described as winter conditions, with snow on the ground present the entire time. Average temperature during the campaign was -7.6°C, with lows down to -23°C during the first part of the campaign and highs up to 4°C towards the end of the campaign. The first half of the campaign was dominated by high-pressure systems, with winds normally from the N-NE-E-SE sectors. The second half of the campaign was dominated by low-pressure systems, with winds normally from the SW-W-NW-N sectors. Median total particle number concentration throughout the campaign was 1996 cm$^{-3}$, with 5$^{th}$ and 95$^{th}$ percentile being 591 cm$^{-3}$ and 5554 cm$^{-3}$, respectively. Of the total 40 measurement days, new particle formation was observed on half of those days (Dal Maso et al., 2005). 59 experiments were conducted with PINC and 393 [INP] values were measured throughout the campaign.

### 3.1 INP number concentration

#### 3.1.1 Data treatment

As mentioned in the "Methodology" section, average [INP] values can be calculated including or excluding values below LOD. Table 1 presents INP number concentrations calculated using both methods. As expected, the median [INP]$_{excl<LOD}$ is higher than the median [INP]$_{incl<LOD}$ at 17.1 and 12.8 L$^{-1}$, respectively. A two-sample $t$-test revealed no significant difference between the two datasets at a 5% confidence interval. This indicates that across the whole field campaign the exclusion of data points below LOD does not lead to a significant overestimation of [INP], and either method of data treatment is appropriate. However, Table 1 also shows the results before and after March 18$^{th}$. The date of March 18$^{th}$ was chosen due to a three-day break in measurements. It can be seen that before March 18$^{th}$ the difference between [INP]$_{excl<LOD}$ and [INP]$_{incl<LOD}$ is larger and is significant at a 5% confidence interval. This is a direct consequence of [INP] values being generally lower before March 18$^{th}$ and, therefore, more values falling below LOD. On the contrary, after March 18$^{th}$ INP concentrations were generally higher, fewer values fell below LOD, and the inclusion/exclusion of values below LOD did not lead to a significant difference. Logically, the exclusion of values below LOD becomes more important as more values fall below LOD, and across all measurements presented here, i.e. at median [INP] above 10 L$^{-1}$, the chosen data treatment is not crucial. However, Boose et al. (2016b) reported that even at median [INP] as low as 2.2 L$^{-1}$, measured with PINC at Jungfraujoch during the winter of 2014 at similar $T$ and RH conditions, the difference between including/excluding data points below LOD was also small. In an attempt to accurately represent ambient INP number concentrations and due to the lack of statistically significant difference between the datasets, the discussion hereafter focuses solely on [INP]$_{incl<LOD}$.





### 3.1.2 General information

As mentioned in the previous paragraph, the median [INP]$_{incl<LOD}$ during the entire campaign was 12.8 L$^{-1}$, and the time series
of INP number concentration during the campaign can be seen in Fig. 2 (left). INP number concentration shows a general
increase throughout the campaign, which is also seen in Table 1. The median [INP] after March 18$^{th}$ was more than 2 times
higher than before March 18$^{th}$ at 22.5 and 9.4 L$^{-1}$, respectively. Overall, [INP] varied over 3 orders of magnitude throughout
the campaign, with the smallest value of 1 L$^{-1}$ measured on February 27$^{th}$, and the highest value of 416 L$^{-1}$ measured on March
25$^{th}$. No diurnal profile or a significant difference between daytime and nighttime [INP] values were found.

Figure 2 (right) shows the normalized frequency distribution of all [INP]$_{incl<LOD}$ values, with the distribution being nearly
lognormal. Such feature has been seen in previous INP studies (Maruyama, 1961; Isaac and Douglas, 1971; Radke et al., 1976;
Welti et al., 2018) and has been attributed to successive random dilutions (Ott, 1990). It can, therefore, be said that the ambient
INP concentrations measured in the boreal environment are a result of the random dilution during transport, and that there are
no local sources of INPs at the measurement location. Similar to the conclusions drawn by Welti et al. (2018) for Cabo Verde,
the measured INP concentrations in the boreal environment are representative of the background conditions influenced mostly
by long-range transport. The postulated absence of local sources, however, still does not answer the question of where the
measured INPs come from and what their composition is.

In an attempt to identify potential sources of the measured [INP] in the boreal environment of southern Finland, a trajectory
analysis was performed using the HYPSLIT_4 (HYbrid Single-Particle Lagrangian Integrated Trajectory) trajectory model
(Draxler and Hess, 1998; Heinzerling, 2004). Each 48-hour back trajectory was calculated with the arrival time corresponding
to the mid-point of the [INP] measurement time. Trajectories were calculated for the arrival height of 100 m. The results are
seen in Fig. 3. Panel A shows some spatial variability in the trajectory-based concentration values. It can be seen that air masses
originating in the east and south of the measurement location are typically associated with low INP concentrations at the
measurement site. At the same time, elevated [INP] values are expected in air masses originating from northern sectors, i.e.
north-east towards the Kola Peninsula and north-west above the Norwegian Sea. In order to assess the potential importance
and significance of these source regions, Fig. 3B shows the number of trajectories passing through each grid cell, and Fig. 3C
shows the fraction of trajectories in each grid cell that pass at an elevation of < 100 m above the ground. It can be seen that
both source areas mentioned above have a considerable number of trajectories originating in these regions, demonstrating
some statistical robustness of the observed elevated trajectory-based INP concentration values. At the same time, however,
Fig. 3C shows that almost none of the trajectories passing these potential source regions are in contact with the ground surface,
rendering potential marine source of the Norwegian Sea and the boreal forest source NE of Hyytiälä as improbable. It must be
concluded that on a regional scale across a six-week measurement campaign no one particular source region of INPs can be
singled out. This means that a) elevated [INP] seen in Fig. 2 (left) can originate from different areas, and b) there is a likelihood
of long-range transport of INPs to the boreal environment in southern Finland from areas outside of those depicted on the maps
in Fig. 3.



### 3.1.3 Comparison to previous field studies

Comparing [INP] values presented here to values measured elsewhere with similar instrumental setups, it can be said that [INP] in a boreal environment is similar to that measured in other parts of the globe. Boose et al. (2016a) measured [INP] with PINC under similar conditions at the Izaña observatory in Tenerife, Spain and reported mean values of 229 and 23 L$^{-1}$ for the summers of 2013 and 2014, respectively. It should be noted, however, that [INP] values at Izaña are frequently affected by the Saharan dust events, a phenomenon unlikely to have a significant impact on [INP] in the boreal environment of southern Finland (Sogacheva et al., 2005). While [INP] values at Hyytiälä and Izaña can be considered comparable in magnitude, the sources of [INP] are different; this topic is addressed in more detail later on in the paper. PINC measurements at the high Alpine station Jungfraujoch in the winter of 2014 resulted in median [INP] values of 2.2 L$^{-1}$ (Boose et al., 2016b), below those reported here. Similar can be said of measurements at Jungfraujoch performed with Horizontal Ice Nucleation Chamber (HINC) in the winters of 2015 and 2016 (Lacher et al., 2017). The study reported median [INP] values below 5 L$^{-1}$ for both winters. Immersion mode measurements of [INP] at 243 K performed at the coastal marine boundary layer in Western Canada reported mean [INP] values of 3–15 L$^{-1}$ (Mason et al., 2015), i.e. similar to the ones measured in the boreal environment in this study. INP number concentrations measured at a similar temperature in the Amazon rainforest (~10 L$^{-1}$; Prenni et al., 2009) and in the ponderosa pine forest in Colorado (~10–50 L$^{-1}$; Tobo et al., 2013) also agree well with those measured in the boreal environment. Most recently, Welti et al. (2018) reported on the [INP] measured in the subtropical maritime boundary layer at the Cabo Verde islands with the Spectrometer for Ice Nuclei (SPIN) chamber under similar measurement conditions. The study reported a wide range of [INP] values at 241 K, spanning $10^0$–$10^3$ L$^{-1}$. While [INP] values measured in Hyytiälä compare well to some of the previously published results from other locations, it is assumed that INP sources in various environments may be different, contributing varying fractions to the total INP number. Since the boreal environment of southern Finland is unlikely to be significantly influenced by mineral dust, it then becomes even more important to identify potential sources of INPs at the SMEAR II location. It has to be reiterated that [INP] measured in the boreal environment was not notably different from [INP] measured in other parts of the world with similar measurement setups. The data collected at SMEAR II further add to the notion that INPs seem to be distributed rather uniformly around the globe, and that in the absence of local sources one is to expect [INP] values to fall within a certain defined range regardless of the measurement location (Kanji et al., 2017; Lacher et al., 2018).

### 3.1.4 INP characterisation

In order to probe the identity of the measured INPs, a multitude of various meteorological and particle data have been used. Figure 4 depicts the Pearson correlation coefficients ($R$) of the linear correlation between the entire time series of measured [INP] and these various parameters. As seen in the figure, all $R$ values fall inside the shaded area (-0.5 < $R$ < 0.5), indicating a weak correlation or no correlation at all. This means that across the six-week measurement campaign, no one single parameter can be used to predict even up to 50% of the observed [INP]. The absence of any significant correlation has several important





implications. First, it suggests that none of the existing parameterisations used to estimate ambient [INP] would reproduce the [INP] measured in the boreal environment in this study. This is true for parameterisations based on particle size (Richardson
et al., 2007; DeMott et al., 2010) and those including the number of fluorescent biological active particles (FBAPs; Tobo et al., 2013). It seems as though other external parameters, those beyond particle size and the biological fraction, may be important for the observed temporal variability in [INP] in the boreal environment during the measurement time. Second, it also indicates that a variety of particle types may act as INPs at different times. Since the campaign lasted for six weeks, any such changes in INP identity would disappear when averaged over the entire measurement period. This issue is addressed in the following
section. Third, the point-to-point variability in [INP] values highlights the necessity for the high time resolution of ambient [INP] measurements. During the campaign the [INP] values were measured with 25–30 minute intervals or even longer, and the [INP] on any given day varied by as much as 1 order of magnitude. For example, on March 26th the lowest and highest [INP] measured 8 hours apart were 11 and 106 $L^{-1}$, respectively. Such variability would not be visible if INP measurements were conducted over several hours with, for example, a high-volume filter sampler.

As mentioned in the "Methodology" section, the PFPC was deployed during the first month of the campaign, and it may potentially provide some insight into the physical characteristics of the sampled ambient INPs. While the PFPC was in use, each icing cycle of PINC resulted in one INP enrichment factor EF, and a total of 38 EF values were collected during the campaign. Similar to what was reported by Lacher et al. (2018), EF values varied between 5 and 35, and, with the PFPC particle enrichment being size dependent (Gute et al., 2019), the variability in EF reflects the variability of air masses and,
thus, the potential size range of particles active as INPs. As mentioned previously, total aerosol concentration factors were determined for the PFPC on several occasions during the campaign, and comparing EF values to these concentration factors can potentially shed light on the size of measured INPs. The median EF during the campaign is 13, and the total aerosol concentration factor of 13 corresponds to the particle mobility diameter of ~300 nm. This may be an early indication that on average during the measurement campaign INPs were mostly around 300 nm in diameter. In fact, particles of this size range
have previously been identified in ice crystal residuals (Mertes et al., 2007; Kupiszewski et al., 2016). Additionally, EF values correlate best with particles in the size range of 100–500 nm ($R = 0.61$), strengthening the possibility that ambient INPs in the boreal environment are likely in this size range.

One important aspect needs to be addressed when correlating ambient INP number concentrations with particle physical and chemical properties. Activated fraction (AF) values, calculated as a ratio of the [INP] to the total number of particles entering
the measurement setup, span 3 orders of magnitude, with a median value of $1.46 \times 10^{-5}$. This median value indicates that, on average, only 1 in $10^5$ ambient aerosol particles acted as an INP. On March 25th, a day with particularly elevated [INP], AF values increased to $1.1 \times 10^{-4}$, meaning that 1 in $10^4$ particles acted as an INP. At the same time measurements with WIBS and L-ToF-AMS were performed in bulk, i.e. sampling the entire aerosol population. The absence of correlations between [INP] and particle chemical properties is, therefore, completely unsurprising. It is to be expected that an instrument measuring in
bulk would not be able to characterise 1 in $10^4$ particles. Same can be said of reasons why [INP] does not exhibit significant correlations with any of the particle size channels despite indications that ambient INPs may be in the 100–500 nm size range.



Not every 300 nm particle will act as an INP, and, since their exact identity remains unknown, one is to expect an absence of correlation between [INP] and total number of particles in the 100–500 nm size range. This has important implications for future ambient INP measurements with CFDCs. If the INP physical and chemical properties are of interest, one needs to capture the activated INPs, i.e. ice crystals, at the exit of the chamber for further offline analysis. Such endeavour can be performed, for instance, with a pumped counterflow virtual impactor (PCVI) (e.g. Corbin et al., 2012).

## 3.2 Special case studies

As mentioned in the previous sections, the campaign lasted for six weeks, and the measurement location experienced various weather conditions and air masses over this time period. Therefore, if one is to assume that different particles act as INPs at different times, their signatures would be lost in the process of averaging over the entire duration of the field campaign. In order to investigate the possible identity of different INPs, the focus is placed on shorter time periods, characterised by particular weather conditions.

### 3.2.1 27–28 February 2018

This time period clearly stood out due to its particularly stable winter conditions. These two days were characterised by the presence of a high-pressure system, with the highest $P$ and lowest $T$ during the entire campaign. The winds were exclusively from the NE sector, and the sky was totally clear. The 48-hour trajectory analysis revealed that air masses were indeed arriving from the north-east, from as far as the Kanin Peninsula (Fig. 5a). The air masses arriving at Hyytiälä during this time period were in contact with the ground during the entire 48 hours prior to arrival. Total particle number concentrations were elevated, with a median of 3526 cm$^{-3}$, while [INP] were lower than the campaign median, at 7.4 L$^{-1}$. A total of 23 [INP] data points were measured during this time period.

Figure 5b presents Pearson correlation coefficients of the linear correlation between the [INP] and various parameters on February 27$^{th}$ and 28$^{th}$. Immediately visible in the figure is that certain parameters exhibit moderate/strong and statistically significant correlation with [INP]. [INP] positively correlates with BC (Fig. 5c), as well as with various particle size channels, mostly below 1 μm in diameter, and total particle surface area. This situation raises the questions of whether BC is able to act as an INP under our measurement conditions during the examined time period and where this BC may be coming from. The trajectories in Fig. 5a show that air masses arriving in Hyytiälä were in contact with the surface before their arrival, with at least 24 last hours spent above the land surface. This means that aerosol particle properties, including those of INPs, in the arriving air masses are likely influenced by the surface emissions of the boreal forest. The contact of air masses with the ground is also reflected in elevated total aerosol particle number. Black carbon is emitted into the atmosphere during the biomass combustion, and the BC particles typically exhibit sizes in the accumulation mode, i.e. below 1 μm in diameter (Reid and Hobbs, 1998). This notion supports the presence of moderate correlations of [INP] with both BC and the total aerosol particle number in sizes below 1 μm in diameter. The source of trajectories and their contact with the ground may suggest that BC is originating from the wood burning and heating of households and saunas in the northwestern Russia and Eastern Finland



during particularly cold weather conditions. Despite the short day length, the cloudless conditions may result in the freshly
emitted BC being subject to photochemical oxidation, leading to an increase in its ability to act as CCN/INP (Li et al., 2018).
Soot is generally regarded as an inefficient INP under mixed-phase cloud conditions (Friedman et al., 2011; Chou et al., 2013;
Mahrt et al., 2018). However, BC from biomass burning has been shown to be IN-active at 243 K and above water saturation
in a previous field study by Prenni et al. (2012), supporting the notion of IN-active BC originating from wood burning and
heating during the examined time period. Additionally, the absence of any other IN-active species may also lead to a clear BC
signature in the INP measurements presented here (Thomson et al., 2018). It seems as though the potential role of BC as INPs
active under mixed-phase cloud regime during these particular weather conditions cannot be excluded.

To conclude, it should be noted that in this case BC may be simply a tracer for some other unknown INP species, the
characteristics of which are not seen in any of the examined variables beyond that of particle size. It is not possible to explicitly
say whether it is BC or some other particle species acting as an INP during this measurement period. Additionally, the absolute
concentration of BC during these two days is similar to the overall average BC concentration. Reasons why the signature of
BC as an INP shows up only during this time period and not others remain unknown.

### 3.2.2 9–11 March 2018

In contrast to the previously described time period, these three days were characterised by the presence of a low-pressure
system, with the air temperature steady at ~ -1.5°C and ambient RH always at or close to 100%. Winds were predominantly
from the southern sector, with fully cloudy conditions and a cloud base height of 100–200 metres above ground level. The 48-
hour trajectory analysis showed that air masses were arriving from the south-south-east (Fig. 6a). Earlier in this time period
the air masses originated in Western Russia, while those arriving in the afternoon of March 11[th] were coming from Estonia. In
all cases, however, the air masses originated mostly above the continental land mass and were in contact with the ground for
most of the time prior to their arrival at the measurement site. Both total particle number concentrations and [INP] were slightly
below average, with median values of 1455 cm$^{-3}$ and 6.6 L$^{-1}$, respectively. A total of 36 [INP] data points were measured
during this time period.

Figure 6b shows that [INP] exhibits a moderate and statistically significant linear correlation with total aerosol numbers at
larger particle sizes. Most notably, [INP] correlates best with the number of particles 1–2.5 μm in diameter and the total particle
surface area (Fig. 6c). This supports the general notion that larger, supermicron particles are better INPs (e.g., Kanji et al.,
2011); although it does not explain why the observed correlation exists only for this certain time period and not for the entire
dataset. [INP] also exhibits a positive correlation with fluorescent concentration, as well as with organics, ammonium and
sulfate as measured by the L-ToF-AMS. Biological fluorescent particles have previously been identified as important INPs
during field studies in both marine locations (Mason et al., 2015) and in forested sites (Tobo et al., 2013). It has also been
reported that a high ambient relative humidity, which is the case during this time period, could trigger the release of biological
particles into the atmosphere (Wright et al., 2014). It can, therefore, be said that significant linear correlations of [INP] with
supermicron and fluorescent particles, as well as with organics, accompanied by the very high ambient RH likely indicate the





importance of these biological particles released by the surrounding forest as INPs. It should be noted, however, that of the 36 [INP] data points during this time period, only 12 have corresponding WIBS concentration measurement. The observed correlation of [INP] with ammonium and sulfate, while unexpected, may be explained by the ability of ammonium sulfate to

increase particle water uptake (Rose et al., 2008), potentially leading to aerosol particles becoming better INPs in the immersion mode. The puzzling aspect of this time period that remains unresolved is the fact that despite the potential release of biological aerosol and the particles' increase in CCN and INP activity due to ammonium sulfate, the absolute [INP] concentration during this time period is lower than average.

### 3.2.3 25 March 2018

March 25th stood out in the entire campaign as the day with the highest measured INP number concentration. During this day a total of 14 consecutive [INP] values were measured, each over a 15-minute period. The lowest and highest [INP] values on March 25th were 52 and 416 L$^{-1}$, respectively, meaning that the day was characterised by unusually high [INP] compared to the campaign average. Times series of [INP] and a multitude of other relevant parameters can be seen in Fig. 7. When the meteorological conditions are examined, it becomes clear that a significant change in weather occurred just before 15:00:00

local time. A cold front passed in the middle of the afternoon, with a warm low-pressure system with above-zero temperatures and winds from the south-east being replaced by a high-pressure system with low, sub-zero temperatures and northerly winds. The time series of [INP] show a gradual increase in the early afternoon, with the peak [INP] value of 416 L$^{-1}$ around 14:00:00 local time, approximately an hour before the frontal passage. INP number concentration then starts decreasing, reaching its minimum value of 52 L$^{-1}$ around the exact time of the frontal passage. [INP] then remained relatively low and stable for the

rest of the day. It must be immediately noted that none of the chemical groups measured by the L-ToF-AMS exhibited any changes due to the cold front passage (Fig. 7). The same can be said of the concentrations of BC and fluorescent biological particles. When the total ambient aerosol population is examined, only the concentration of particles below 100 nm in diameter responded to a change in weather, similar to that of [INP]. Moreover, when the correlation of [INP] with particles of smallest sizes on March 25th is examined (not shown), the Pearson correlation coefficients $R$ were above 0.75 for the correlation of

[INP] with both particles below 0.01 μm and particles in the 0.01–0.1 μm size range. What this might mean is that INPs on this day are below 0.1 μm in diameter, potentially even below 0.01 μm in diameter. The absence of correlation with mass spectrometry groups and biological fluorescent particles then becomes obvious as WIBS only measures particles over 0.5 μm in diameter, and the low size for 100% transmission efficiency of the L-ToF-AMS is about 0.045 μm with most of the smaller particles likely being lost in the transmission lens. It is, therefore, not possible to speculate about the chemical composition of

these sub-0.1 μm particles that are most likely acting as INPs on this particular day.
The new particle formation (NPF) analysis revealed that 25 March 2018 is classified as *undefined*, meaning that the day could not be classified as either an event or a non-event (Dal Maso et al., 2005). Figure 8a shows the time series of the particle number size distribution as measured by the local DMPS. The figure reveals that there is a growing Aitken mode visible between 13:00:00 and 15:00:00 local time. Nucleation mode particles are also present; however, the concentration of Aitken



mode particles starts increasing earlier in the day than the concentration of sub-0.01 μm particles (Fig. 7). This indicates that NPF is not a likely source of sub-0.1 μm particles on this day. The particle size distribution then undergoes a dramatic change around 15:00:00 local time due to the frontal passage, with the number of Aitken mode particles dramatically decreasing. Figure 8b shows the origins of the air masses arriving at the measurement site on March 25th. As expected, air masses arriving before the frontal passage originated in the south west, as far as Denmark and Southern Sweden. After the frontal passage air

masses originated in Central Finland. In both cases for most of the time prior to their arrival, air masses were in contact with the ground. Despite a significant change in weather and air mass origin, none of the ambient particle physical and chemical properties, except number concentration of particles below 0.1 μm in diameter, changed as a result of the frontal passage. The source of the sub-0.1 μm particles visible in Fig. 8a just prior to the frontal passage remains uncertain. As mentioned above, local NPF is not likely a source of these particles. And even if it were, freshly formed organic aerosol, especially at such small

sizes, is unlikely to exhibit such increased ice nucleation activity at the RH and $T$ conditions being sampled by PINC (Möhler et al., 2008; Kanji et al., 2017). At the same time, long-range transport is also unlikely as the Aitken mode particles only appeared around noon local time despite the fact that air mass origin was stable for many hours prior to the frontal passage. One possible identity of these highly IN-active sub-0.1 μm particles could be that of nanoscale biological fragments, such as ice nucleating macromolecules INMs. INMs and other nanoscale biological fragments have previously been observed to be

IN-active (e.g. Govindarajan and Lindow, 1988; Hartmann et al., 2013; O'Sullivan et al., 2015), and these fragments can be as small as 0.01 μm in size (Kanji et al., 2017). Sources of these sub-0.1 μm biological particles can be pollen (Pummer et al., 2012), fungal spores (Fröhlich-Nowoisky et al., 2015) and soil dust (O'Sullivan et al., 2015), although it needs to be mentioned that on March 25th the ground at the measurement site was still covered with snow, rendering soil dust an unlikely source of the INMs. Similarly, it may be unreasonable to expect high pollen and fungal activity at the measurement location in March

given the meteorological conditions during the examined day (Fig. 7). However, the figure shows that prior to the arrival of the cold front, ambient temperature was above freezing and increasing, and the speculation here is that the above-zero temperatures would trigger at least some biological activity in the boreal forest, which could potentially result in the emission of the IN-active, sub-0.1 μm biological fragments. In fact, March 25th had the highest ambient temperature during the entire campaign, making this hypothesis plausible. Given the available data, it seems as though the highest measured [INP] during

the campaign can be attributed to highly IN-active nanoscale biological fragments originating from surrounding vegetation. There are two questions that remain unanswered within the realm of the current hypothesis. First, in the early afternoon hours of March 25th the total concentration of particles below 100 nm in diameter was approximately 2000 cm⁻³, which is very similar to the median total particle number concentration across all sizes of 1996 cm⁻³ during the campaign. Assuming no NPF, ~2000 cm⁻³ of sub-0.1 μm particles is significant, and it is unknown whether biological activity of the surrounding boreal forest in the

sub-5°C temperature regime can be responsible for such a high concentration. Although, as mentioned above, March 25th was indeed the warmest day during the entire campaign. Second, and similar to other special cases examined, it remains unknown why the importance of IN-active sub-0.1 μm biological fragments is only visible on this day and not during other periods or during the entire campaign. This question would be answered if the release of these biological fragments from biota and the



increased [INP] would be observed during other warmer periods with temperatures above 4°C. However, March 25th was the
only day during the campaign when this temperature was exceeded.

**4 Conclusion**

The median INP number concentration measured at 242 K and 105% $RH_w$ during a six-week measurement campaign in
southern Finland was found to be 13 L$^{-1}$. The measured [INP] spanned 3 orders of magnitude and showed a general increase
from mid-February until early April. No statistically significant difference was found between [INP] with values below LOD
excluded and included, indicating the insignificance of the chosen data treatment method of PINC ambient measurements at
[INP] > 10 L$^{-1}$. No persistent local or regional sources of INPs in the boreal environment of southern Finland could be
identified. Instead, it is postulated that the INPs at SMEAR II are a result of dilution during long-range transport. Despite high
variability, the measured [INP] values fall within the range expected for INP number concentrations measured elsewhere at
similar thermodynamic conditions. [INP] did not correlate with any of examined relevant parameters during the entire field
campaign, indicating that no one single parameter can be used to predict the INP number concentration at the measurement
location during the examined time period. The absence of correlation across the entire field campaign also suggests that a
variety of particles are acting at INPs at different times, although it was indirectly determined that, on average, ambient INPs
are most likely in the size range of 0.1–0.5 μm in diameter. This result highlighted the necessity for the high time resolution
of INP measurements, as any measurements averaged across several hours would erase the variability in INP identity. On
shorter time scales, several particle species correlated well with [INP] implying their potential role as INPs. Depending on the
meteorological conditions, signatures of BC and supermicron biological particles have been found in the INP signal. However,
the increase in BC or in supermicron biological particle may not necessarily lead to the increase in [INP], reasons for which
remain unknown. On the day with the highest [INP], sub-0.1 μm particles, most likely nanoscale biological fragments such as
INMs, were found to significantly contribute to the elevated INP number concentrations. Reasons for why certain particle
types act as INPs during certain conditions and not during others and why none of the particle species mentioned above
correlate with [INP] across the entire campaign remain unknown.

The main results summarized above present several important questions and conclusions for future ambient INP studies. First
and foremost, a running theme throughout the paper is the inability to identify physical and chemical properties of ambient
INPs despite the complexity of instrumental setup and a multitude of additional instrumentations deployed during the
campaign. Unfortunately, the current instrumental setup coupled with the rarity of ambient INPs rendered any speculations
about INP identity inconclusive. This could have been foreseen as INPs are very rare, and all additional instrumentation
measured aerosol properties in bulk. If the future INP field studies aim to explicitly identify the physical and chemical
properties of INPs, a single-particle analysis of ice crystal residuals must be carried out, similar to the works of Mertes et al.
(2007), Richardson et al. (2007), Corbin et al. (2012) and Schmidt et al. (2017). Moreover, in order to establish particle
properties responsible for its ability to act as INP, physics and chemistry of the total aerosol population must be examined as



well. The second important question that needs to be addressed as a result of this work is the necessity for future INP measurements. As mentioned earlier, the measured [INP] values fell within the range expected from previous field studies in different environments around the globe (Kanji et al., 2017). While it may be important to quantify [INP] in various environments around the world, the [INP] values themselves present little information about their potential interactions with

clouds and the resulting ice crystal number concentration. This notion is further supported by the fact that aerosol properties on the ground are quite likely different from those found at altitudes where examined thermodynamic conditions may actually be met. It, therefore, needs to be said that INP measurements similar to the ones presented here are of limited use in the studies of aerosol-cloud interactions. Future ambient INP studies need to explicitly focus on the physical and chemical properties of the INPs and whether they are different from the rest of the aerosol population. Additionally, future studies should aim to

assess whether the explicit physics and chemistry of INPs can be and need to be parameterised in order to understand aerosol-cloud interactions on the global scale.

**Competing interests**

The authors declare that they have no conflict of interest.

**Acknowledgements**

This project has received funding from the European Union's Horizon 2020 research and innovation programme under the Marie Sklodowska-Curie grant agreement No 751470 "ATM-METFIN". The research leading to these results has also received funding from the European Union Seventh Framework Programme (FP7/2007-2013) under grant agreement n° 262254. This project has also received funding from the European Union's Horizon 2020 research and innovation programme under grant agreement No 654109. XC received funding from the European Regional Development Fund, Project MOBTT42. PH and JK

received funding from the Arctic Academy Programme "ARKTIKO" of Academy of Finland, project grant No. 286558 "ICINA". EG and JPDA were supported by NSERC (Grant Number RGPIN-2017-05972), and EG received support from the University of Toronto Centre for Global Change Science Graduate Student Research Award. Liqing Hao was funded by the European Research Council (Starting grant 335478). Liine Heikkinen received funding from ERC-Stg COALA.

Olga Garmash and Zoé Brasseur are thanked for their help with the L-ToF-AMS measurements. Pasi Aalto and Petri Keronen

are thanked for the timely provision of DMPS/APS and meteorological data, respectively. Simo Hakala is acknowledged for his help with the new particle formation classification. Zoé Brasseur and Jonathan Duplissy are both acknowledged for their involvement with the campaign activities. The authors would also like to gratefully acknowledge the staff of Hyytiälä Forestry Field Station. Their expertise, hard work and willingness to help have made the campaign a great success! Larissa Lacher, Robert David, Fabian Mahrt and Dimitri Castarède are all sincerely thanked for the helpful discussions and continuous

encouragement. Prof. Ulrike Lohmann is acknowledged for her support and guidance.



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





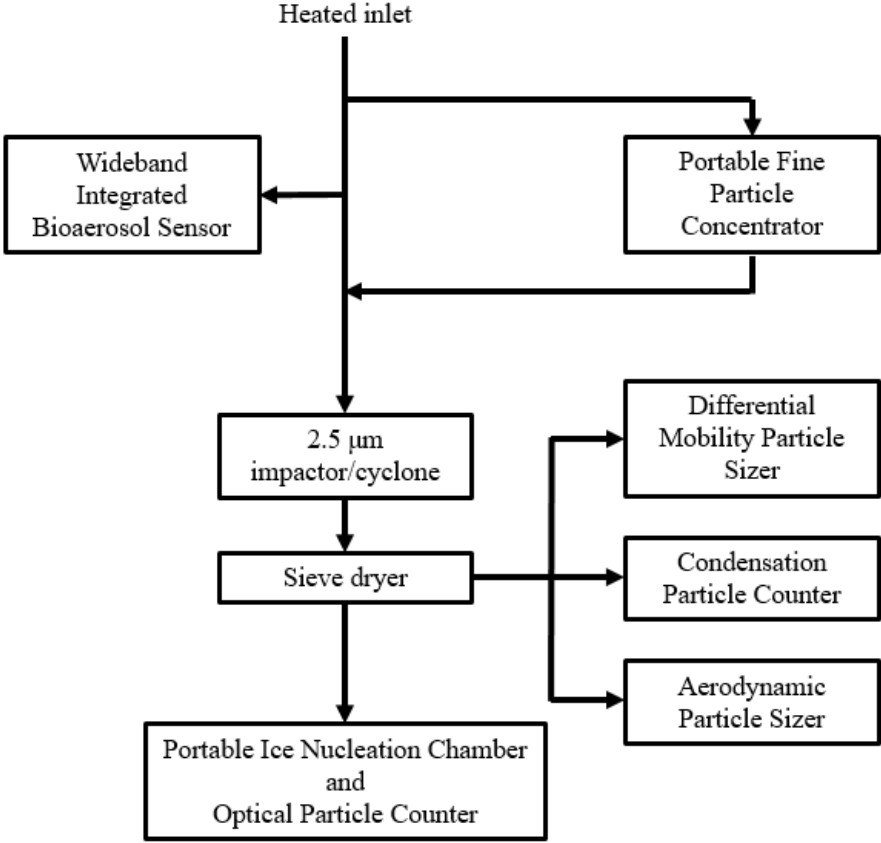

**Figure 1: Instrumental setup. All other relevant instrumentation was located < 100 m away from the main setup; see text for details.**







**Table 1: Median, 5th and 95th percentile values of the INP number concentration calculated by including/excluding values below LOD.**

| | $[INP]_{incl<LOD}$ (L⁻¹) | | | $[INP]_{excl<LOD}$ (L⁻¹) | | |
|---|---|---|---|---|---|---|
| | 5th percentile | median | 95th percentile | 5th percentile | median | 95th percentile |
| **all data** | 4.2 | 12.8 | 86.0 | 6.9 | 17.1 | 86.0 |
| **before March 18th** | 3.8 | 9.4 | 37.0 | 6.5 | 14.7 | 47.2 |
| **after March 18th** | 5.3 | 22.5 | 132.0 | 8.4 | 23.9 | 132.0 |




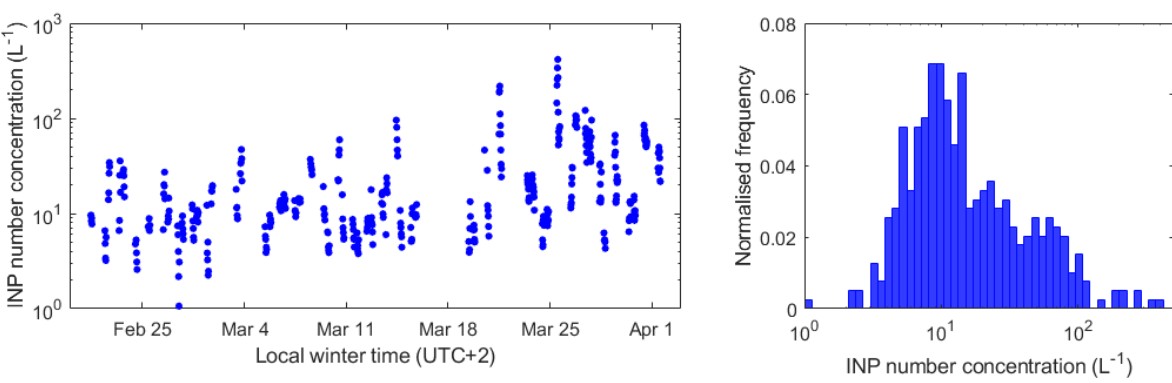


**Figure 2: [INP] time series throughout the whole campaign (left) and its normalised frequency distribution (right).**





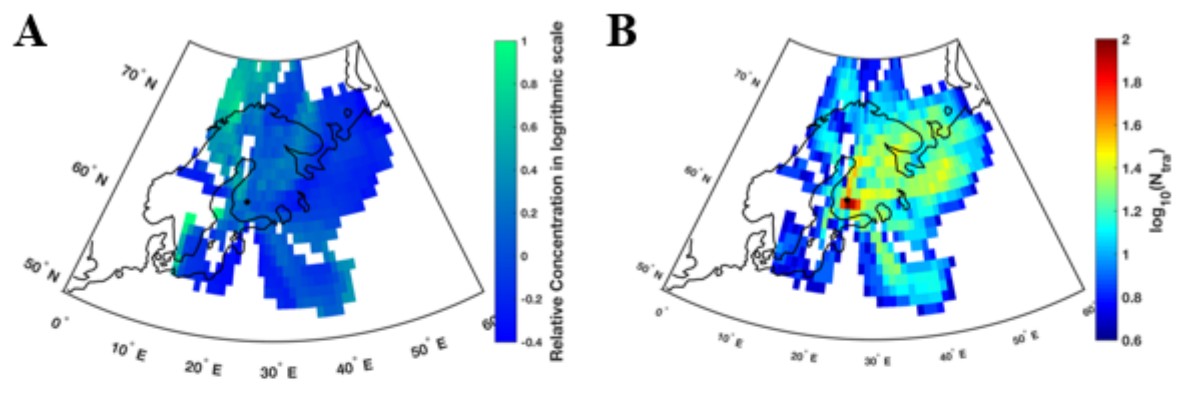

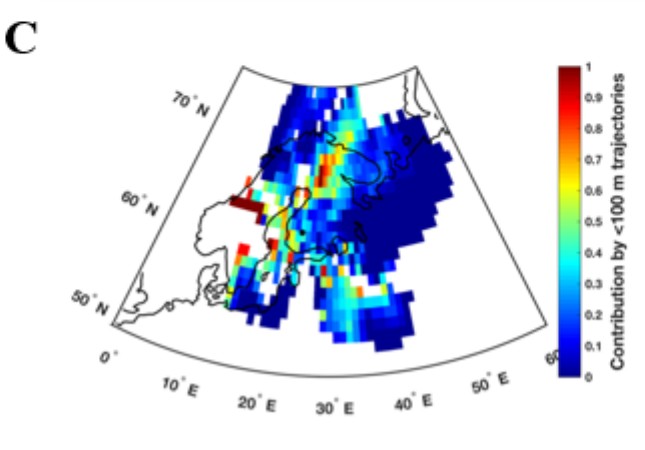

**Figure 3: Panel A: Trajectory-based frequency plot of [INP] across the entire campaign. The concentration fields are shown on a logarithmic scale, and the colour bar indicates the deviation from the measured median INP number concentration of 12.8 L$^{-1}$ (zero on the colour bar). Panel B: Number of trajectories passing through each grid cell. Panel C: Fraction of trajectories shown in panel B that pass at an elevation of < 100 m above the ground. In all panels the grid size is 1°x1°, and only those grid cells with at least five trajectories passing through them are shown. The location of the Hyytiälä Forestry Field Station is marked with a black dot.**





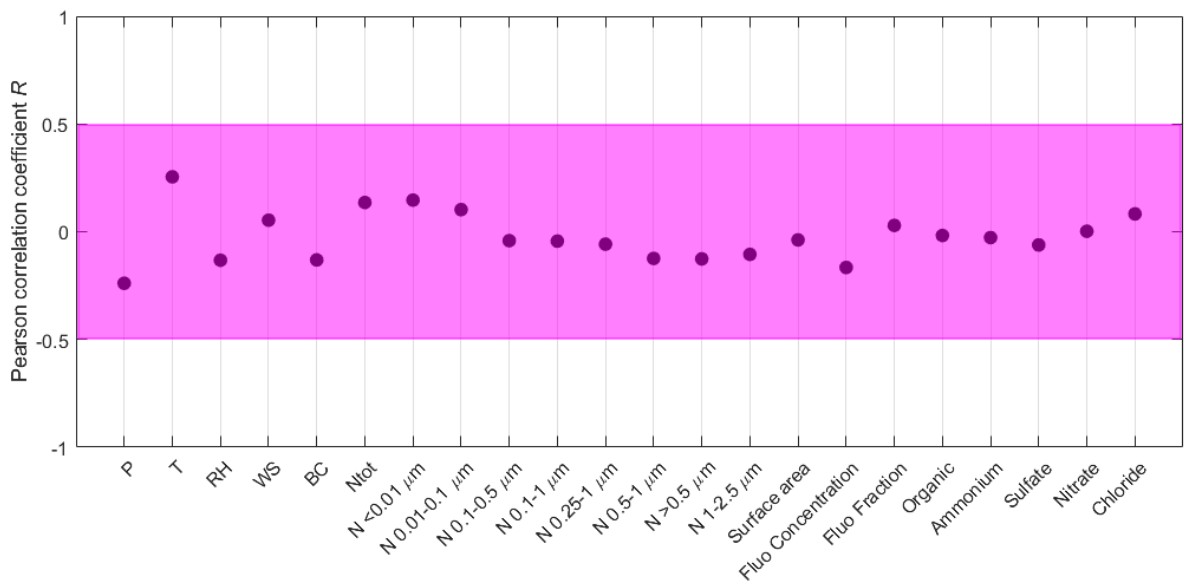


**Figure 4: Pearson correlation coefficients of the linear correlation between the entire time series of [INP] and various meteorological and particle parameters. Purple shading denotes the area of weak, if any, correlation (-0.5 < *R* < 0.5).**





**Figure 5: All results shown for 27–28 February 2018. Panel A: Backward trajectories calculated with HYSPLIT model, extending**
**48 hours back in time. Panel B: Pearson correlation coefficients of the linear correlation between [INP] and various meteorological and particle parameters. Purple shading denotes the area of weak, if any, correlation (-0.5 < R < 0.5). Yellow shading denotes parameters for which the correlation with corresponding [INP] is better than weak (R > 0.5) and statistically significant at 95% confidence interval (p-value < 0.05). Neither WIBS, nor L-ToF-AMS were running yet during this time period. Panel C: [INP] as a function of BC concentration.**






**Figure 6: All results shown for 9–11 March 2018. Panel A: Backward trajectories calculated with HYSPLIT model, extending 48 hours back in time. Panel B: Pearson correlation coefficients of the linear correlation between [INP] and various meteorological and particle parameters. Purple shading denotes the area of weak, if any, correlation (-0.5 < R < 0.5). Yellow shading denotes parameters for which the correlation with corresponding [INP] is better than weak (R > 0.5) and statistically significant at 95% confidence**
**interval (p-value < 0.05). Panel C: [INP] as a function of total particle surface area.**





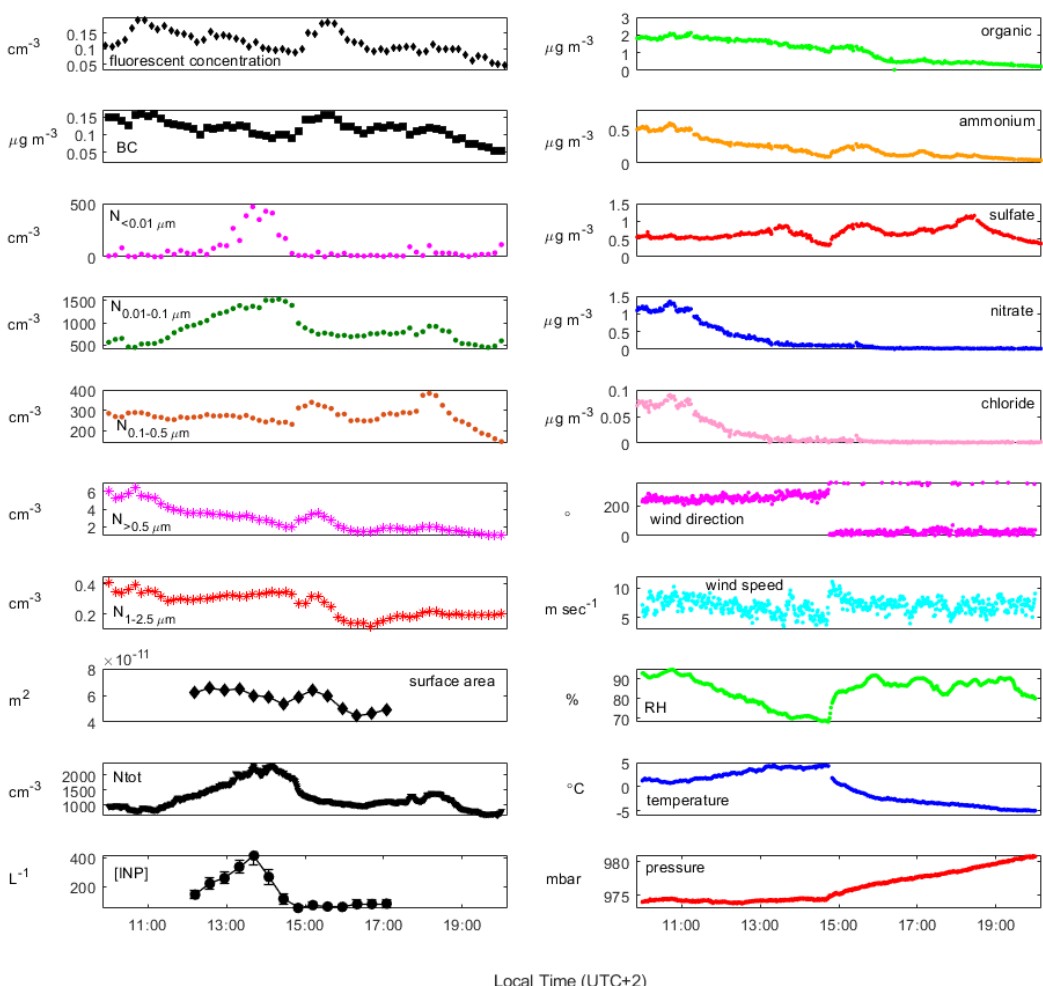

**Figure 7: Time series of [INP] and other relevant parameters on 25 March 2018. Each panel is labelled with the parameter it shows, and the corresponding units are located adjacent to the relevant axis.**





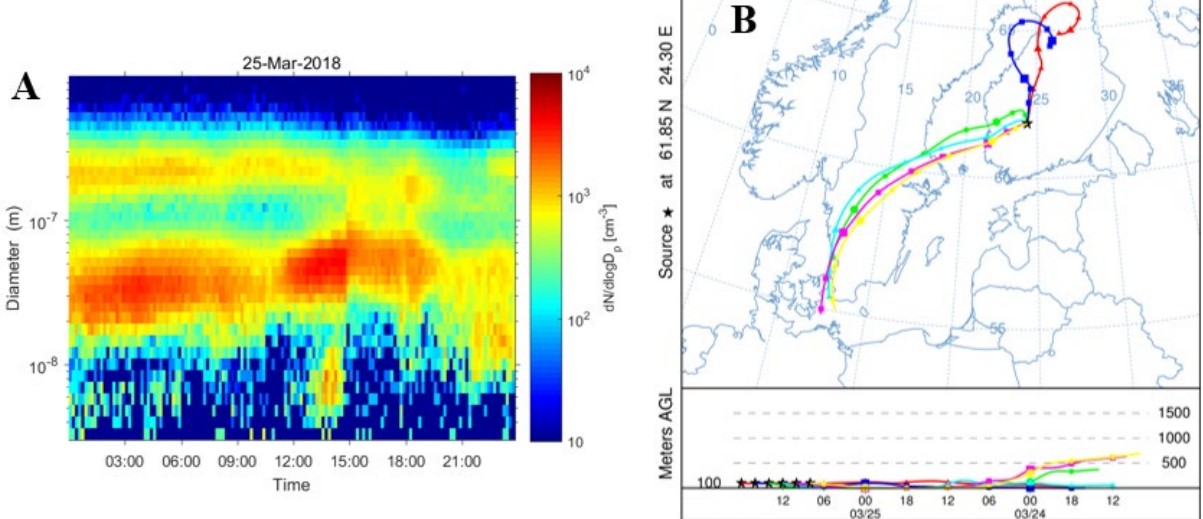

**Figure 8: Panel A: Time series of the particle number size distribution as measured by the local DMPS. Panel B: Backward trajectories calculated with HYSPLIT model, extending 48 hours back in time. Both panels depict 25 March 2018.**