# Peer review of "Condensation/immersion mode ice nucleating particles in a boreal environment"

_Atmospheric Chemistry and Physics, 2019_

## Referee Comment (RC1) · Anonymous Referee #1 · 10 Oct 2019

**Review of "Condensation/immersion mode ice nucleating particles in a boreal environment"**

Anonymous Reviewer

October 10, 2019

**1 Summary**

In this work, Paramonov et al. report the first measured, condensation/immersion-mode INP concentrations from a boreal forest. Measurements were taken from 19 February to 2 April 2018 in Hyytiälä, Finland. INP concentrations were measured using the PINC instrument; on March 2nd, an L-TOF-AMS co-sampled with PINC, and a WIBS was added to both PINC and the L-TOF-AMS on March 11. In addition, total aerosol concentrations (CPC), size distributions (DMPS, APS), and black carbon (MAAP) were measured at the nearby SMEAR II station.

These measurements are important, as they provide constraints to INP concentrations from models in a region where few INP measurements exist. They also provide some insight into the natural variability of INP at a single site over several months.

That being said, the authors are extremely liberal in their interpretations. This is especially true in Section 3.2. The authors are INP experts, thus this reviewer is surprised that Pearson correlation coefficients are being used to chemically speciate the measured INP. As the authors note several times in the paper, INP are a rare subset of all aerosol particles. Thus, correlation certainly does not imply causation, and much of the paper beyond Section 3.1.2 suffers from this questionable-cause logical fallacy.

I have outlined my primary concerns in the general comments section.

In addition, the language in the manuscript is generally wordy and often imprecise. Although this is a stylistic concern, it is prevalent enough throughout the manuscript that it detracts from its clarity. It is the responsibility of all authors on the manuscript to ensure that the manuscript language is clear, and the results are evident. As an example, I will provide minor/technical comments on the abstract paragraph; however, similar corrections need to be applied to the entire manuscript.

**2  General Comments**

1. Section 2: As it is written as one, large chunk of text, the methods section is hard to digest. It would be helpful to the reader if the authors split this section into more subsections, (*e.g.*, PINC, instruments co-located with PINC, instruments located at SMEAR II station, back trajectory analysis, etc.). I also could not find much detail about the back trajectory analysis in the methods section.

2. Figure 1: Figure 1 is missing the L-TOF-AMS. It is also unclear from the figure that the DMPS, CPC, and APS were not at the same location as PINC, WIBS, and the L-TOF-AMS.

3. Line 286: I am not sure why the authors believe that the back trajectories render a "potential marine source of the Norwegian Sea and the boreal forest source NE of Hyytiälä as improbable." To show surface influence, it would be more correct to see how many back trajectories were below the boundary layer height. Presumably, even during NH, high-latitude winter/spring, aerosol are well-mixed within the boundary layer. I strongly suggest that the back trajectory analysis be changed from <100 m to the <boundary layer height.

4. Figure 3: It is odd that a low fraction of trajectories at the SMEAR II site do not encounter the surface. Is this a known feature of Hyytiälä in the NH winter/spring? Or is this a result of your back trajectory arrival height? If the latter, did the authors test how sensitive the back trajectory results are to arrival height? Without such a sensitivity analysis–it would be even more difficult to exclude local INP sources from marine regions and the nearby boreal forests.

5. Section 3.1.3: I do not see the value of this section; the range of INP concentrations in Hyytiälä at one temperature spanned almost 3 orders of magnitude. As CFDCs have lower detection limits around 1 $\mathrm{L}^{-1}$, then this indicates that these measurements are similar to any site where the [INP] spans $\sim$1-1000 $\mathrm{L}^{-1}$.

6. Section 3.2: As stated in the summary section, I find no reason why high correlations with the 22 measurements in this manuscript suggests anything about the composition of the actual INP. Thus, I do not believe that these correlations alone implicate BC, large biological particles, or small, biological nanoparticles as the INPs measured by PINC.

7. Section 3.2.1: There is very little evidence in the literature that BC acts as an immersion-mode INP at activated fractions relevant to this work. This is true for both fossil fuel emissions (Schill et al., 2016) and for biomass burning surrogates (Levin et al., 2016), whose BC INP activated fractions are $sim$1x10$^{-9}$. It has also been shown that photochemical aging does not increase the INP efficiency of BC (Schill et al., 2016), in contradiction to

the statement made on line 390. Thus, the INPs responsible are likely not BC, but some INP co-emitted with BC and present in activated fractions of $\sim$2x10$^{-6}$. Furthermore, although this becomes a focal point of this section, there is little observational evidence that this BC is from residential heating. Finally, BC is not the only measurement that correlated with INP. INP are positively correlated with almost all of the aerosol indicators >100 nm, suggesting that there is something special about the air mass, not the BC, that is supplying INPs. One last specific note–the reference to the Prenni et al. (2012) paper is incorrect. They find INP in biomass burning emissions, but they do not attribute them to BC specifically. A follow-up paper by McCluskey et al. (2014) does show that BC can be found in INP residuals while sampling from prescribed burns, but the BC INP activated fractions were not reported.

8. Section 3.2.2: Similar to BC in Section 3.2.1, This section does not implicate biological particles as INP. Again, the authors ignore that INPs are correlated with all particles >100 nm, and focus only on a subset of their observations. Thus, again, the correlations indicate that a certain air mass type is correlated with [INP], not that a certain type of aerosol are INP.

9. Section 3.2.3: I am not sure why the authors chose to show the time series here instead of a figure similar to Figures 5 and 6; however, even without the Pearson correlation coefficients, I agree with the authors that the correlation with sub-100-nm particles is striking. The authors suggest that the INP must also be sub-100-nm–this, however, is not supported by any observations. The authors hypothesize that these sub-100-nm INP biological nanoparticles, likely because most other known INP lose their ice nucleation activity below 100 nm (Marcolli et al., 2007). To support this hypothesis, the authors note that biological nanoparticles have previously been implicated as INP (Pummer et al., 2012; Fröhlich-Nowoisky et al., 2015; O'Sullivan et al., 2014); however, these biological nanoparticles are found in the ambient atmosphere attached to carrier particles >100 nm such as pollen, fungal spores, and soil dust. The ice nucleating entities were determined to be <100 nm by rinsing the ice nucleating entities off of pollen, soil dust, etc. To the reviewer's knowledge, no study has observed unattached, ice nucleation active, sub-100-nm biological nanoparticles in ambient aerosol samples. Thus, attributing biological nanoparticles as the INP responsible for the high-[INP] event on 25 March 2018 is speculative at best.

**3   Stylistic Concerns: Abstract Example**

- Line 15: Parentheses around SMEAR II

- Line 17: Delete the phrase "found to be," it is wordy and slightly redundant

- Line 18: The INPs are not necessarily "a result of" dilution and long range transport. This suggests that dilution and long-range transport create INPs. The INPs are a result of long-range transport and dilution of INPs sourced far from the measurement site. This needs to be clear.

- Line 21: You already made an abbreviation for INP number concentrations ([INP])–please use it here.

- Line 23: The phrase "any of the examined relevant parameters," is vague here. If parameters do not correlate, then are they relevant? Furthermore, since you are not using these parameters to define INPs (or any system), they are not parameters. They should be called measurements or observations.

- Line 24: Again you have already abbreviated INP number concentrations to [INP].

- Line 25: You use the subordinating conjunction "although," which suggests that you should omit the comma beforehand. In fact, "although" is connecting two independent ideas–thus, it would be clearer for the reader if you split this sentence in two.

- Line 28: You should not connect "correlated" with "found in," because they are not the same thing. The former is true, you did find a correlation; the latter is not, you did not find anything in the INP.

**References**

Fröhlich-Nowoisky, J., Hill, T. C., Pummer, B. G., Yordanova, P., Franc, G. D., and Pöschl, U.: Ice nucleation activity in the widespread soil fungus Mortierella alpina, Biogeosciences, 12, 1057–1071, https://doi.org/10.5194/bg-12-1057-2015, 2015.

Levin, E. J. T., McMeeking, G. R., DeMott, P. J., McCluskey, C. S., Carrico, C. M., Nakao, S., Jayarathne, T., Stone, E. A., Stockwell, C. E., Yokelson, R. J., and Kreidenweis, S. M.: Ice-nucleating particle emissions from biomass combustion and the potential importance of soot aerosol, Journal of Geophysical Research: Atmospheres, 121, 5888–5903, https://doi.org/10.1002/2016JD024879, URL http://doi.wiley.com/10.1002/2016JD024879, 2016.

Marcolli, C., Gedamke, S., Peter, T., and Zobrist, B.: Efficiency of immersion mode ice nucleation on surrogates of mineral dust, Atmospheric Chemistry and Physics, 7, 5081–5091, https://doi.org/10.5194/acp-7-5081-2007, 2007.

McCluskey, C. S., DeMott, P. J., Prenni, A. J., Levin, E. J. T., McMeeking, G. R., Sullivan, A. P., Hill, T. C. J., Nakao, S., Carrico, C. M., and Kreidenweis, S. M.: Characteristics of atmospheric ice nucleating particles associated with biomass burning in the US: Prescribed burns and wildfires, Journal of Geophysical Research: Atmospheres, 119, 10 458–10 470, https://doi.org/10.1002/2014JD021980, URL `http://doi.wiley.com/10.1002/2013JG002552` `http://doi.wiley.com/10.1002/2014JD021980`, 2014.

O'Sullivan, D., Murray, B. J., Malkin, T. L., Whale, T. F., Umo, N. S., Atkinson, J. D., Price, H. C., Baustian, K. J., Browse, J., and Webb, M. E.: Ice nucleation by fertile soil dusts: Relative importance of mineral and biogenic components, Atmospheric Chemistry and Physics, 14, 1853–1867, https://doi.org/10.5194/acp-14-1853-2014, 2014.

Prenni, A. J., Demott, P. J., Sullivan, A. P., Sullivan, R. C., Kreidenweis, S. M., and Rogers, D. C.: Biomass burning as a potential source for atmospheric ice nuclei: Western wildfires and prescribed burns, Geophysical Research Letters, 39, 1–5, https://doi.org/10.1029/2012GL051915, 2012.

Pummer, B. G., Bauer, H., Bernardi, J., Bleicher, S., and Grothe, H.: Suspendable macromolecules are responsible for ice nucleation activity of birch and conifer pollen, Atmospheric Chemistry and Physics, 12, 2541–2550, https://doi.org/10.5194/acp-12-2541-2012, 2012.

Schill, G. P., Jathar, S. H., Kodros, J. K., Levin, E. J., Galang, A. M., Friedman, B., Link, M. F., Farmer, D. K., Pierce, J. R., Kreidenweis, S. M., and DeMott, P. J.: Ice-nucleating particle emissions from photochemically aged diesel and biodiesel exhaust, Geophysical Research Letters, 43, 5524–5531, https://doi.org/10.1002/2016GL069529, 2016.

---

## Referee Comment (RC2) · Anonymous Referee #2 · 30 Oct 2019

**Review of manuscript "Condensation/immersion mode ice nucleating particles in a boreal environment" by Mikhail Paramonov and coauthors**

Paramonov et al. studied the ice nucleating particles (INPs) in the condensation/immersion mode in the boreal environment of southern Finland during winter-spring of 2018. The number concentrations of INPs were measured using a continuous flow diffusion chamber PINC, along with the measurements of total aerosol particles (DMPS, CPC, APS), aerosol chemical composition (L-ToF-AMS), biological fluorescent particles (WIBS), and meteorological conditions (RH, T, WS, etc.). The measurements were used to investigate the number concentrations, sources, and possible compositions of INPs at this location during the studied time period. A few case studies were also presented to show the variability of physical and chemical properties of INPs over a short time period. This study is important as it is a nice addition to the rare INP measurements conducted in the boreal forest environment, and will help improve our understanding of INPs in the atmosphere. However, some of the conclusions/hypothesises reached in the manuscript were not well supported by the data, along with some other issues that the authors may consider to address in the revision.

Major comments:

1. P1 L18 and P9 L268: the conclusion "there are no local sources of INPs" cannot be drawn based solely on the lognormal distribution of [INP] frequency. Welti et al. (2018) only suggested "the absence of a strong local source". Also, this conclusion is contradictory to the fact that biological particles released by the surrounding forest were considered as a source of INPs in Sect. 3.2.2. Please revise the statement to make it clear.

2. P1 L25: The conclusion "ambient INPs are most likely in the size range of 0.1-0.5 µm in diameter" was not well supported by the data. Fig.4 shows that overall INPs didn't correlate with $N_{0.1-0.5\mu m}$ at all. Also, the design of the setup removed all the particles >2.5 µm, which may contribute to a large fraction of INPs (Mason et al., 2016). This should be discussed in the manuscript.

3. Introduction: most of the result discussion focused on the composition and size information of INPs. Corresponding background information about compositions and sizes of INPs should be expanded in the introduction.

4. Sect. 2.2: it's hard to navigate through this section. Subsections of each instrument or instrument type are recommended. Also it's confusing what instruments are in operation at different time of the campaign (e.g. PFPC, L-ToF-AMS, WIBS), a table listing the operation time period of each instrument might be helpful. When using "the first half and the second half of the campaign", please specify what period is considered as first half and what period is considered as the second half.

5. The last paragraph on P9, a few comments regarding the back trajectories: (1) L276: arrival height of 100m above ground level or sea level? The site is 181m a.m.s.l and the inlet is 2m tall. Why doesn't the arrival height match the height of the inlet? Are the trajectories sensitive to the height? (2) L280: For people who don't know the geography of Europe very well, it's hard to tell which area you're referring to by saying "north-east towards the Kola Peninsula and north-west above the Norwegian Sea". Please add labels on the map, or include a separate map panel.

6. Sect 3.2.2: it has been mentioned that the surrounding ground has been covered by snow, how about the area where the air masses come from? Was it covered by snow as well during the campaign? Would mineral dust be a possible source? I agree with the author that the correlation with fluorescent particles made the biological particles a likely source. But the mineral dust particles can't be fully ruled out.

Minor comments:

7. P2 L55 and P3 L75: the discussions of [INPs] in the atmosphere are redundant.

8. P4 L119: typo "dryer"

9. P8 L245: was it 5% confidence interval or 95%? If 5%, is it reasonable to compare two data sets at a 5% confidence interval?

10. P9 L274: typo "HYSPLIT"

11. Section 3.1.3: the authors should be careful when comparing INP measurements. The size range of INPs, the techniques could be different. For example, Mason et al. (2015) measured INPs using a different technique than PINC and measured INPs up to 10μm.

12. Fig.4: how are the size ranges determined? It seems a little bit random. There are some overlapping. Also, does $N_{tot\ >0.5\mu m}$ mean $N_{tot\ 0.5-2.5\mu m}$?

13. P11, L352: a recent paper (Si et al., 2018) correlated the activation fraction with the INP size, which supports your observation here.

14. Fig. 5 and 6: the capital letters A, B, C are used in the figures, while lower cases a, b, c are used in the text.

15. Fig. 5A and 6A: how are the back trajectories generated? Does the arrival time still correspond to the mid-point of the INP measurement time? It seems like a new trajectory was generated every 6h.

**References:**

Mason, R. H., Si, M., Li, J., Chou, C., Dickie, R., Toom-Sauntry, D., Pöhlker, C., Yakobi-Hancock, J. D., Ladino, L. A., Jones, K., Leaitch, W. R., Schiller, C. L., Abbatt, J. P. D., Huffman, J. A. and Bertram, A. K.: Ice nucleating particles at a coastal marine boundary

layer site: correlations with aerosol type and meteorological conditions, Atmos. Chem. Phys., 15(21), 12547–12566, doi:10.5194/acp-15-12547-2015, 2015.

Mason, R. H., Si, M., Chou, C., Irish, V. E., Dickie, R., Elizondo, P., Wong, R., Brintnell, M., Elsasser, M., Lassar, W. M., Pierce, K. M., Leaitch, W. R., MacDonald, A. M., Platt, A., Toom-Sauntry, D., Sarda-Estève, R., Schiller, C. L., Suski, K. J., Hill, T. C. J., Abbatt, J. P. D., Huffman, J. A., DeMott, P. J. and Bertram, A. K.: Size-resolved measurements of ice-nucleating particles at six locations in North America and one in Europe, Atmos. Chem. Phys., 16(3), 1637–1651, doi:10.5194/acp-16-1637-2016, 2016.

Si, M., Irish, V. E., Mason, R. H., Vergara-Temprado, J., Hanna, S. J., Ladino, L. A., Yakobi-Hancock, J. D., Schiller, C. L., Wentzell, J. J. B., Abbatt, J. P. D., Carslaw, K. S., Murray, B. J. and Bertram, A. K.: Ice-nucleating ability of aerosol particles and possible sources at three coastal marine sites, Atmos. Chem. Phys., 18(21), 15669–15685, doi:10.5194/acp-18-15669-2018, 2018.

Welti, A., Müller, K., Fleming, Z. L. and Stratmann, F.: Concentration and variability of ice nuclei in the subtropical maritime boundary layer, Atmos. Chem. Phys., 18(8), 5307–5320, doi:10.5194/acp-18-5307-2018, 2018.

---

## Author Comment (AC1) · 10 Feb 2020

**Review of "Condensation/immersion mode ice nucleating particles in a boreal environment"**
**Anonymous Reviewer**
**October 10, 2019**

**1. Summary**
**In this work, Paramonov et al. report the first measured, condensation/immersion mode INP concentrations from a boreal forest. Measurements were taken from 19 February to 2 April 2018 in Hyytiälä, Finland. INP concentrations were measured using the PINC instrument; on March 2nd, an L-TOF-AMS co-sampled with PINC, and a WIBS was added to both PINC and the L-TOF-AMS on March 11. In addition, total aerosol concentrations (CPC), size distributions (DMPS, APS), and black carbon (MAAP) were measured at the nearby SMEAR II station.**

**These measurements are important, as they provide constraints to INP concentrations from models in a region where few INP measurements exist. They also provide some insight into the natural variability of INP at a single site over several months.**

Authors' response: The authors would like to thank the reviewer for a detailed and insightful review. The line numbers below refer to the version without track changes.

**That being said, the authors are extremely liberal in their interpretations. This is especially true in Section 3.2. The authors are INP experts, thus this reviewer is surprised that Pearson correlation coefficients are being used to chemically speciate the measured INP. As the authors note several times in the paper, INP are a rare subset of all aerosol particles. Thus, correlation certainly does not imply causation, and much of the paper beyond Section 3.1.2 suffers from this questionable-cause logical fallacy.**

Authors' response: The authors acknowledge the limitations of using the Pearson correlation coefficients to interpret the data. The instrumentation setup during the field campaign did not allow for the single-particle analysis of ice crystal residuals, and this is addressed on lines 537-541 (page 17). Additionally, the fact that correlation *does not* imply causation is caveated throughout the manuscript on several occasions, e.g. lines 413-417 (page 13-14). This manuscript is not the first attempt at linearly correlating [INP] with various parameters in order to be able to say something about the physical and chemical properties of ambient INPs. There is a multitude of studies that have derived some use from this approach (Tobo et al., 2013; Mason et al., 2015; Wright et al., 2015; Welti et al., 2018).

The authors further note that Pearson correlation coefficients are not used to chemically speciate the measured INP. They are used to infer the predicting capacity of [INP], i.e. to explore if a parameter (and which one) can be used to predict the ambient INP number concentration. Predicting [INP] is one of the biggest challenges facing the IN communities, and attempts to find such parameter have been proposed before (Richardson et al., 2007; DeMott et al., 2010; Tobo et al., 2013). However, we add the following clarifications to address the reviewer concerns regarding correlation and causality.

- Lines 444-448 (pages 14-15), added text "It should also be noted here that correlation of [INP] with fluorescent particle concentration does not imply that INPs are necessarily fluorescent. Similar to the previous case, fluorescent particles may simply be a feature of the present airmass or a tracer for some other IN-active particle species. However, it is possible that fluorescent particles do contribute to the INP population given previous studies that show INP residuals sampled from the atmosphere to show an increased fluorescent fraction compared to background measurements (Boose et al., 2016).".
- Lines 510-512 (page 16), added text "Third, and also similar to other special cases examined, a significant correlation between [INP] and the concentration of sub-0.1 μm particles does not imply that sub-0.1 μm particles are INPs; instead, they may be a tracer of some IN-active particle species or an airmass feature.".
- Line 530-531 (page 17), sentence modified to read "On the day with the highest [INP], sub-0.1 μm particles, most likely nanoscale biological fragments such as INMs, were found to exhibit a significant correlation with the elevated INP number concentrations."

**I have outlined my primary concerns in the general comments section.**

**In addition, the language in the manuscript is generally wordy and often imprecise. Although this is a stylistic concern, it is prevalent enough throughout the manuscript that it detracts from its clarity. It is the responsibility of all authors on the manuscript to ensure that the manuscript language is clear, and the results are evident. As an example, I will provide minor/technical comments on the abstract paragraph; however, similar corrections need to be applied to the entire manuscript.**

Authors' response: We acknowledge the stylistic concerns of the reviewer. However, some redundancy is intentional to in order to remind and guide the reader through the manuscript. However, the language of the manuscript has undergone some modification (see tracked changes version) to account for reviewer's comments.

**2. General comments**
1. **Section 2: As it is written as one, large chunk of text, the methods section is hard to digest. It would be helpful to the reader if the authors split this section into more subsections, (e.g., PINC, instruments co-located with PINC, instruments located at SMEAR II station, back trajectory analysis, etc.). I also could not find much detail about the back trajectory analysis in the methods section.**
   Authors' response: Section 2.2 is now split into several subsections as per reviewer's suggestion. A new subsection 2.2.6 is added to describe the trajectory analysis, and the related text moved from section 3.1.2 to section 2.2.6.

2. **Figure 1: Figure 1 is missing the L-TOF-AMS. It is also unclear from the figure that the DMPS, CPC, and APS were not at the same location as PINC, WIBS, and the L-TOF-AMS.**
   Authors' response: Figure 1 has been updated to include all instrumentation that was used, and the figure caption updated to read: "Figure 1: Instrumental setup.".

3. **Line 286: I am not sure why the authors believe that the back trajectories render a "potential marine source of the Norwegian Sea and the boreal forest source NE of Hyytiälä as improbable." To show surface influence, it would be more correct to see how many back trajectories were below the boundary layer height. Presumably, even during NH, high-latitude winter/spring, aerosol are well-mixed within the boundary layer. I strongly suggest that the back trajectory analysis be changed from <100 m to the <boundary layer height.**
   Authors' response:
   - We choose the trajectory arrival height of 100 m because we are interested in the potential influence of the Earth's surface and its emissions.
   - The definition of the boundary layer implies a well-mixed layer that is influenced by the surface. So if the BL height is 1 km, the trajectory arrival height should not matter so long as it is within the 1 km layer. This height should be well-representative of the BL conditions at SMEAR II. Only if the BL height was below 100 m would the trajectory arrival height of 100 m be inappropriate for studying the effects of the surface. But this is unlikely since most of the INP measurements presented took place during the day when the BL, as implied by the reviewer comments, is > 100 m.

4. **Figure 3: It is odd that a low fraction of trajectories at the SMEAR II site do not encounter the surface. Is this a known feature of Hyytiälä in the NH winter/spring? Or is this a result of your back trajectory arrival height? If the latter, did the authors test how sensitive the back trajectory results are to arrival height? Without such a sensitivity analysis-it would be even more difficult to exclude local INP sources from marine regions and the nearby boreal forests.**
   Authors' response:
   - Actually, Figure 3 shows the contrary. With a few exceptions, a very low fraction of trajectories is in contact with the surface. This is signified by the prevalence of blue colours in Figure 3C. While it is not possible to say if this is a known feature at SMEAR II, we did look at the results when the trajectory arrival height was set to 200 m and 500 m. In both of these cases, the likelihood of contact with the ground was even less, and the potential sources of INPs became even more inconclusive. This is consistent with expectations. If the airmasses arriving at a height of 100 m come from above 100 m, it is not very likely that airmasses arriving at a height of 500 m would be coming from below 100 m.
   - Since we are interested in the surface effects, the lowest possible arrival height of 100 m seems most appropriate. No changes made.

5. **Section 3.1.3: I do not see the value of this section; the range of INP concentrations in Hyytiälä at one temperature spanned almost 3 orders of magnitude. As CFDCs have lower detection limits around 1 L$^{-1}$, then this indicates that these measurements are similar to any site where the [INP] spans ~1-1000 L$^{-1}$.**
   Authors' response:
   - Our measurements and results are not stand-alone; they are part of an overall endeavour by the IN community to characterise ambient INPs. It is, therefore, necessary to provide context, to see how our results fit in the bigger picture and

how they compare to previous studies. Thus, though the value of this section may not be apparent, we keep this section in for the above reason and good scientific practice of comparing new results to those published previously on a similar topic.

- The reviewer is correct in saying that "the range of INP concentrations in Hyytiälä at one temperature spanned almost 3 orders of magnitude". The reviewer is also correct in saying that "these measurements are similar to any site where the [INP] spans ~1-1000 L-1", which is basically any site around the world without strong local sources. This section is meant to explicitly highlight that the range of ambient [INP] seems to be same around the globe and, given this, to raise the question of whether similar ambient INP measurements are necessary in the future. See lines 543-547 (page 17-18). No changes made.

6. **Section 3.2: As stated in the summary section, I find no reason why high correlations with the 22 measurements in this manuscript suggests anything about the composition of the actual INP. Thus, I do not believe that these correlations alone implicate BC, large biological particles, or small, biological nanoparticles as the INPs measured by PINC.**
Authors' response: This issue is addressed above in the summary section. Each of the 3.2 sub-sections includes an explanation that the particle types correlating with [INP] may not actually be the particles acting as INPs. This can be seen on lines 413-417 (page 13-14), lines 444-446 (page 14), lines 469-470 (page 15) and lines 510-512 (page 16).

7. **Section 3.2.1: There is very little evidence in the literature that BC acts as an immersion-mode INP at activated fractions relevant to this work. This is true for both fossil fuel emissions (Schill et al., 2016) and for biomass burning surrogates (Levin et al., 2016), whose BC INP activated fractions are sim1x10$^{-9}$. It has also been shown that photochemical aging does not increase the INP efficiency of BC (Schill et al., 2016), in contradiction to the statement made on line 390. Thus, the INPs responsible are likely not BC, but some INP co-emitted with BC and present in activated fractions of ~2x10$^{-6}$.**
Authors' response: Sentence in lines 405-406 (page 13) has been modified to read "Despite the short daylength, the cloudless conditions may result in the freshly emitted BC being subject to photochemical oxidation, leading to an increase in its ability to act as CCN (Li et al., 2018)." See further responses below.

**Furthermore, although this becomes a focal point of this section, there is little observational evidence that this BC is from residential heating.**
Authors' response:

- It has been reported in previously published literature that the concentration of BC at SMEAR II is highest in February and March (Virkkula et al., 2011). The study used the light absorption technique to come to this conclusion. At the same time, Lewis et al. (2008) provided evidence that high light absorption coefficients are typically associated with different types of biomass burning. It, therefore, becomes quite likely that the source of BC at SMEAR II in February and March is biomass burning.

- Sentence added in lines 403-405 (page 13): "This notion has been indirectly supported by Lewis et al. (2008) and Virkkula et al. (2011) who reported that the high aerosol light absorption at SMEAR II in February and March is likely associated with biomass burning."

**Finally, BC is not the only measurement that correlated with INP. INP are positively correlated with almost all of the aerosol indicators >100 nm, suggesting that there is something special about the air mass, not the BC, that is supplying INPs.**
Authors' response: Yes, that's correct. It is well-known that large particles are more likely to act as INPs than smaller ones (Hoose and Möhler, 2012; Pruppacher and Klett, 1997). However, size alone does not tell us anything about what these INPs could be. We have found an interesting correlation with BC and we investigated further using existing literature. Whether BC is acting as an INP or is simply a tracer/airmass feature remains open for further research as mentioned in the manuscript (lines 413-417, pages 13-14).

**One last specific note-the reference to the Prenni et al. (2012) paper is incorrect. They find INP in biomass burning emissions, but they do not attribute them to BC specifically. A follow-up paper by McCluskey et al. (2014) does show that BC can be found in INP residuals while sampling from prescribed burns, but the BC INP activated fractions were not reported.**
Authors' response: Sentence in lines 407-410 (page 13) modified to read: "However, biomass burning has been shown to produce INPs active above water saturation at 243 K (Prenni et al. 2012) and warmer temperatures (McCluskey et al., 2014), supporting the notion of IN-active biomass burning particles originating from wood burning and heating during the examined time period.".

8. **Section 3.2.2: Similar to BC in Section 3.2.1, This section does not implicate biological particles as INP. Again, the authors ignore that INPs are correlated with all particles >100 nm, and focus only on a subset of their observations. Thus, again, the correlations indicate that a certain air mass type is correlated with [INP], not that a certain type of aerosol are INP.**
   Authors' response:
   - Lines 444-448 (pages 14-15), added text "It should also be noted here that correlation of [INP] with fluorescent particle concentration does not imply that INPs are necessarily fluorescent. Similar to the previous case, fluorescent particles may simply be a feature of the present airmass or a tracer for some other IN-active particle species. However, it is possible that fluorescent particles do contribute to the INP population given previous studies that show INP residuals sampled from the atmosphere to show an increased fluorescent fraction compared to background measurements (Boose et al., 2016).".
   - We agree with the reviewer in that the correlation in this subsection indicates that a certain air mass type is correlated with [INP]; however, this implies that the INP correlate with aerosol in the air mass, given INP are a subset of condensed phase aerosols and not gas phase species. It is possible that the gas phase species in an air mass serve to modify the aerosol species causing an enhancement or

suppression of [INP]; however, the correlation should still be with the aerosol properties within an air mass type.

9. **Section 3.2.3: I am not sure why the authors chose to show the time series here instead of a figure similar to Figures 5 and 6;**
Authors' response: We wanted the reader to see the evolution of [INP] on what we think is the most interesting day of our measurements, as well as to have a visual cue as to what was happening on that day with respect to the weather and other parameters. Figures 5 and 6 do not provide such information.

**however, even without the Pearson correlation coefficients, I agree with the authors that the correlation with sub-100-nm particles is striking. The authors suggest that the INP must also be sub-100-nm-this, however, is not supported by any observations.**
Authors' response: Addressed above and in lines 413-417 (page 13-14), 444-448 (pages 14-15) and 510-512 (page 16). Also, see response below regarding sub-100-nm particles.

**The authors hypothesize that these sub-100-nm INP biological nanoparticles, likely because most other known INP lose their ice nucleation activity below 100 nm (Marcolli et al., 2007). To support this hypothesis, the authors note that biological nanoparticles have previously been implicated as INP (Pummer et al., 2012; Fröhlich-Nowoisky et al., 2015; O'Sullivan et al., 2014); however, these biological nanoparticles are found in the ambient atmosphere attached to carrier particles >100 nm such as pollen, fungal spores, and soil dust. The ice nucleating entities were determined to be <100 nm by rinsing the ice nucleating entities off pollen, soil dust, etc.**
Authors' response: That's correct. Previous studies have found IN-active biological nanoparticles attached to larger, carrier particles. However, if pollen particles during the warmest day of the measurement campaign are to rupture (either due to high humidity or mechanical rupture from turbulent wind conditions), sub pollen particles (SPPs), which are in the Aitken mode, are released and thus suspended in air on their own. It is reported that up to 70% of pollen grains rupture in the troposphere and release SPPs which get dispersed into the troposphere (Wozniak et al., 2018 and references therein). It is not necessary for biological fragments to be appended by carrier dust or soil particles. Previous studies have shown that biological molecules can be transported on soil or desert dust to long distances, but this is not the only form of bio-molecules existing in the atmosphere, and certainly not for the boundary layer, relevant to our measurement site. Thus, we cannot neglect the likelihood of these particles existing independently at least for some non-trivial amount of time, especially close to the emission source.

**To the reviewer's knowledge, no study has observed unattached, ice nucleation active, sub-100-nm biological nanoparticles in ambient aerosol samples. Thus, attributing biological nanoparticles as the INP responsible for the high-[INP] event on 25 March 2018 is speculative at best.**
Authors' response: Yes, we agree. However, the fact that SPPs have been observed to be active CCN even in the Aitken mode (Steiner et al., 2015 GRL) suggests that it is very

likely these particles can act as immersion INPs. Furthermore, most particle detection techniques used to identify ice crystal residuals are limited to observing populations of aerosols > 100 nm (cite Cziczo et al., 2017, AMS monographs), which could be a reason for this. However, the correlation here, albeit not a causation, can demonstrate the importance and need to consider potential identities of sub-100 nm INPs.

**3. Stylistic Concerns: Abstract Example**

- **Line 15: Parentheses around SMEAR II**
  corrected here and in line 103

- **Line 17: Delete the phrase "found to be," it is wordy and slightly redundant**
  corrected here and in line 515

- **Line 18: The INPs are not necessarily "a result of" dilution and long range transport. This suggests that dilution and long-range transport create INPs. The INPs are a result of long-range transport and dilution of INPs sourced far from the measurement site. This needs to be clear.**
  corrected here and in lines 518-520 to say "…INPs at SMEAR II are a result of long-range transport and dilution of INPs sourced far from the measurement site".

- **Line 21: You already made an abbreviation for INP number concentrations ([INP])-please use it here.**
  corrected

- **Line 23: The phrase "any of the examined relevant parameters," is vague here. If parameters do not correlate, then are they relevant? Furthermore, since you are not using these parameters to define INPs (or any system), they are not parameters. They should be called measurements or observations.**
  The world "relevant" removed here and in lines 453, 522, and 909.

- **Line 24: Again you have already abbreviated INP number concentrations to [INP].**
  corrected

- **Line 25: You use the subordinating conjunction "although," which suggests that you should omit the comma beforehand. In fact, "although" is connecting two independent ideas-thus, it would be clearer for the reader if you split this sentence in two.**
  The conjunction "although" is used here instead of "but" or "however", and it is meant to split two independent clauses. Using "Although" at the beginning of the sentence would make a fragment sentence. No changes made.

- **Line 28: You should not connect "correlated" with "found in," because they are not the same thing. The former is true, you did find a correlation; the latter is not, you did not find anything in the INP.**
  corrected here and in lines 527-528. "Signatures of" also removed in both instances.

**References**

- Fröhlich-Nowoisky, J., Hill, T. C., Pummer, B. G., Yordanova, P., Franc, G. D., and Pöschl, U.: Ice nucleation activity in the widespread soil fungus Mortierella alpina, Biogeosciences, 12, 1057-1071, https://doi.org/10.5194/bg-12-1057-2015, 2015.
- Levin, E. J. T., McMeeking, G. R., DeMott, P. J., McCluskey, C. S., Carrico, C. M., Nakao, S., Jayarathne, T., Stone, E. A., Stockwell, C. E., Yokelson, R. J., and Kreidenweis, S. M.: Ice nucleating particle emissions from biomass combustion and the potential importance of soot aerosol, Journal of Geophysical Research: Atmospheres, 121, 5888-5903, https://doi.org/10.1002/2016JD024879, URL http://doi.wiley.com/10.1002/2016JD024879, 2016.
- Marcolli, C., Gedamke, S., Peter, T., and Zobrist, B.: Efficiency of immersion mode ice nucleation on surrogates of mineral dust, Atmospheric Chemistry and Physics, 7, 5081-5091, https://doi.org/10.5194/acp-7-5081-2007, 2007.
- McCluskey, C. S., DeMott, P. J., Prenni, A. J., Levin, E. J. T., McMeeking, G. R., Sullivan, A. P., Hill, T. C. J., Nakao, S., Carrico, C. M., and Kreidenweis, S. M.: Characteristics of atmospheric ice nucleating particles associated with biomass burning in the US: Prescribed burns and wildfires, Journal of Geophysical Research: Atmospheres, 119, 10 458{10 470, https://doi.org/10.1002/2014JD021980, URL http://doi.wiley.com/10.1002/2013JG002552 http://doi.wiley.com/10.1002/2014JD021980, 2014.
- O'Sullivan, D., Murray, B. J., Malkin, T. L., Whale, T. F., Umo, N. S., Atkinson, J. D., Price, H. C., Baustian, K. J., Browse, J., and Webb, M. E.: Ice nucleation by fertile soil dusts: Relative importance of mineral and biogenic components, Atmospheric Chemistry and Physics, 14, 1853-1867, https://doi.org/10.5194/acp-14-1853-2014, 2014.
- Prenni, A. J., Demott, P. J., Sullivan, A. P., Sullivan, R. C., Kreidenweis, S. M., and Rogers, D. C.: Biomass burning as a potential source for atmospheric ice nuclei: Western wildfires and prescribed burns, Geophysical Research Letters, 39, 1-5, https://doi.org/10.1029/2012GL051915, 2012.
- Pummer, B. G., Bauer, H., Bernardi, J., Bleicher, S., and Grothe, H.: Suspendable macromolecules are responsible for ice nucleation activity of birch and conifer pollen, Atmospheric Chemistry and Physics, 12, 2541-2550, https://doi.org/10.5194/acp-12-2541-2012, 2012.
- Schill, G. P., Jathar, S. H., Kodros, J. K., Levin, E. J., Galang, A. M., Friedman, B., Link, M. F., Farmer, D. K., Pierce, J. R., Kreidenweis, S. M., and DeMott, P. J.: Ice-nucleating particle emissions from photochemically aged diesel and biodiesel exhaust, Geophysical Research Letters, 43, 5524-5531,https://doi.org/10.1002/2016GL069529, 2016.

Authors' response references
- Boose, Y., Sierau, B., García, M. I., Rodríguez, S., Alastuey, A., Linke, C., Schnaiter, M., Kupiszewski, P., Kanji, Z. A., and Lohmann, U.: Ice nucleating particles in the Saharan Air Layer, Atmos. Chem. Phys., 16, 9067–9087, doi:10.5194/acp-16-9067-2016, 2016a.
- DeMott, P. J., Prenni, A. J., Liu, X., Kreidenweis, S. M., Petters, M. D., Twohy, C. H., Richardson, M. S., Eidhammer, T., and Rodgers, D. C.: Predicting global atmospheric ice

nuclei distributions and their impacts on climate, P. Natl. Acad. Sci. USA, 107, 11217–11222, 2010.

- Hoose, C., and Möhler, O.: Heterogeneous ice nucleation on atmospheric aerosols: a review of results from laboratory experiments, Atmos. Chem. Phys., 12, 9817–9854, 2012.

- Kanji, Z. A., Ladino, L. A., Wex, H., Boose, Y., Burkert-Kohn, M., Cziczo, D. J., and Krämer, M.: Overview of ice nucleating particles, Meteor. Mon., 58, 1.1–1.33, doi:10.1175/AMSMONOGRAPHS-D-16-0006.1, 2017.

- Lewis, K., Arnott, W. P., Moosmüller, H., and Wold, C. E.: Strong spectral variation of biomass smoke light absorption and single scattering albedo observed with a novel dual-wavelength photoacoustic instrument, J. Geophys. Res., 113, D16203, doi:10.1029/2007JD009699, 2008.

- Mason, R. H., Si, M., Li, J., Chou, C., Dickie, R., Toom-Sauntry, D., Pöhlker, C., Yakobi-Hancock, J. D., Ladino, L. A., Jones, K., Leaitch, W. R., Schiller, C. L., Abbatt, J. P. D., Huffman, J. A., and Bertram, A. K.: Ice nucleating particles at a coastal marine boundary layer site: correlations with aerosol type and meteorological conditions, Atmos. Chem. Phys., 15, 12547−12566, https://doi.org/10.5194/acp-15-12547-2015, 2015.

- Pruppacher, H. and Klett, J.: Microphysics of clouds and precipitation, Atmospheric and oceanographic sciences library; v. 18, Kluwer Academic Publishers, Dordrecht; London, includes bibliographical references and index Previous ed.: 1978 "With an introduction to cloud chemistry and cloud electricity.", 1997.

- Richardson, M. S., DeMott, P. J., Kreidenweis, S. M., Cziczo, D. J., Dunlea, E. J., Jimenez, J. L., Thomson, D. S., Ashbaugh, L. L., Borys, R. D., Westphal, D. L., Casuccio, G. S., and Lersch, T. L.: Measurements of heterogeneous ice nuclei in the western United States in springtime and their relation to aerosol characteristics, J. Geophys. Res., 112, D02209, doi:10.1029/2006JD007500, 2007.

- Tobo, Y., Prenni, A. J., DeMott, P. J., Huffman, J. A., McCluskey, C. S., Tian, G., Pöhlker, C., Pöschl, U., and Kreidenweis, S. M.: Biological aerosol particles as a key determinant of ice nuclei populations in a forest ecosystem, J. Geophys. Res.-Atmos., 118, 10100–10110, doi:10.1002/jgrd.50801, 2013.

- Virkkula, A., Backman, J., Aalto, P. P., Hulkkonen, M., Riuttanen, L., Nieminen, T., dal Maso, M., Sogacheva, L., de Leeuw, G., and Kulmala, M.: Seasonal cycle, size dependencies, and source analyses of aerosol optical properties at the SMEAR II measurement station in Hyytiälä, Finland, Atmos. Chem. Phys., 11, 4445–4468, https://doi.org/10.5194/acp-11-4445-2011, 2011.

- Welti, A., Müller, K., Fleming, Z. L., and Stratmann, F.: Concentration and variability of ice nuclei in the subtropical maritime boundary layer, Atmos. Chem. Phys., 18, 5307–5320, https://doi.org/10.5194/acp-18-5307-2018, 2018.

- Wozniak, M. C., Solmon, F., and Steiner, A. L.: Pollen rupture and its impact on precipitation in clean continental conditions, Geophys. Res. Lett., 45, 7156–7164. https://doi.org/10.1029/2018GL077692, 2018.

- Wright, T. P., Hader, J. D., McMeeking G. R., and Petters, M. D.: High relative humidity as a trigger for widespread release of ice nuclei, Aerosol Sci. Tech., 48, i–v, doi:10.1080/02786826.2014.968244, 2014.

---

## Author Comment (AC2) · 10 Feb 2020

**Review of manuscript "Condensation/immersion mode ice nucleating particles in a boreal environment" by Mikhail Paramonov and coauthors**

**Paramonov et al. studied the ice nucleating particles (INPs) in the condensation/immersion mode in the boreal environment of southern Finland during winter-spring of 2018. The number concentrations of INPs were measured using a continuous flow diffusion chamber PINC, along with the measurements of total aerosol particles (DMPS, CPC, APS), aerosol chemical composition (L-ToF-AMS), biological fluorescent particles (WIBS), and meteorological conditions (RH, T, WS, etc.). The measurements were used to investigate the number concentrations, sources, and possible compositions of INPs at this location during the studied time period. A few case studies were also presented to show the variability of physical and chemical properties of INPs over a short time period. This study is important as it is a nice addition to the rare INP measurements conducted in the boreal forest environment, and will help improve our understanding of INPs in the atmosphere. However, some of the conclusions/hypothesises reached in the manuscript were not well supported by the data, along with some other issues that the authors may consider to address in the revision.**

Authors response: The authors would like to thank the reviewer for a detailed and insightful review. The line numbers below refer to the version without track changes.

**Major comments:**

1. **P1 L18 and P9 L268: the conclusion "there are no local sources of INPs" cannot be drawn based solely on the lognormal distribution of [INP] frequency. Welti et al. (2018) only suggested "the absence of a strong local source". Also, this conclusion is contradictory to the fact that biological particles released by the surrounding forest were considered as a source of INPs in Sect. 3.2.2. Please revise the statement to make it clear.**

   Authors' response: We have reworded this conclusion in lines 18 and 286 to say that "there are no single dominant local sources of INPs".

2. **P1 L25: The conclusion "ambient INPs are most likely in the size range of 0.1-0.5 μm in diameter" was not well supported by the data. Fig.4 shows that overall INPs didn't correlate with $N_{0.1-0.5\mu m}$ at all. Also, the design of the setup removed all the particles >2.5 μm, which may contribute to a large fraction of INPs (Mason et al., 2016). This should be discussed in the manuscript.**
   - The statement that ambient INPs are most likely in the size range of 0.1-0.5 μm in diameter does not stem from the overall correlations shown in Fig.4 The reviewer is correct – we did not find an overall correlation of [INP] with this size channel. Reasons for this are discussed in the last paragraph of the section 3.1.4, i.e. not every 300 nm particle would act as an INP, so a correlation would be unlikely given the low AF values. The conclusion about INPs being 0.1-0.5 μm in size was drawn from the comparison of the INP enrichment factors and total aerosol enrichment factors as measured by the PFPC. This is described in lines 349-361 (page 12).

- The reviewer is also correct that we did not probe INPs over 2.5 µm in diameter. Reasons for excluding particles over 2.5 µm in diameter are discussed in lines 123-125 (page 4).

3. **Introduction: most of the result discussion focused on the composition and size information of INPs. Corresponding background information about compositions and sizes of INPs should be expanded in the introduction.**
Authors' response: The discussion about the size and chemistry of potential INP species has been expanded; see lines 63-71.

4. **Sect. 2.2: it's hard to navigate through this section. Subsections of each instrument or instrument type are recommended. Also it's confusing what instruments are in operation at different time of the campaign (e.g. PFPC, L-ToF-AMS, WIBS), a table listing the operation time period of each instrument might be helpful. When using "the first half and the second half of the campaign", please specify what period is considered as first half and what period is considered as the second half.**
Authors' response:
- Section 2.2 now includes subsections.
- The authors believe that a table describing operation of instruments would be of limited use only. Operating times of the instruments are mentioned in the Methodology section. Figure 1 was updated to include additional instrumentation deployed in the campaign.
- What is meant by the first/second periods of the campaign has now been clarified. See lines 125-126 and 250-253.

5. **The last paragraph on P9, a few comments regarding the back trajectories:**
**(1) L276: arrival height of 100m above ground level or sea level? The site is 181m a.m.s.l and the inlet is 2m tall. Why doesn't the arrival height match the height of the inlet? Are the trajectories sensitive to the height?**
Authors' response:
- Arrival height of trajectories was clarified by modifying the sentence in lines 205-206 to read: "Trajectories were calculated for the arrival height of 100 m above ground level.".
- We selected the arrival height of 100 m agl because we wanted to investigate the effect of surface emissions, and this height is the lowest arrival height for which HYSPLIT can calculate backward trajectories. We also looked at the results when the trajectory arrival height was set to 200 m and 500 m. The trajectories were not sensitive to the arrival height.

**(2) L280: For people who don't know the geography of Europe very well, it's hard to tell which area you're referring to by saying "north-east towards the Kola Peninsula and north-west above the Norwegian Sea". Please add labels on the map, or include a separate map panel.**
Authors' response: Using the text in lines 290-294 together with the Fig. 3A should make it rather clear to the reader where the mentioned areas are. We prefer not to clutter the already busy figure with the labels, but we included clarifications in lines 290-294 to help guide the reader.

6. **Sect 3.2.2: it has been mentioned that the surrounding ground has been covered by snow, how about the area where the air masses come from? Was it covered by snow as well during the campaign? Would mineral dust be a possible source? I agree with the author that the correlation with fluorescent particles made the biological particles a likely source. But the mineral dust particles can't be fully ruled out.**
   Authors' response:
   - It is not possible to quantitatively say whether all the areas where the 48-hour trajectories travelled from were covered by snow. However, looking at this area in Figure 3 and remembering that the campaign took place in February and March, it is quite likely that there was snow on the ground.
   - We do not suspect dust to be a possible source of INPs at the measurement site. We did not find a correlation with larger size bins across the entire campaign, and we did not see trajectories originating from the dust source areas (Figure 3). Mineral dust and its effect on shorter time scales cannot be completely ruled out; however, its presence is unlikely.

**Minor comments:**

7. **P2 L55 and P3 L75: the discussions of [INPs] in the atmosphere are redundant.**

   The first instance highlights the maximum observed [INP] and compares that value to the typical CCN concentrations. The second instance highlights the range of observed [INP] values. Therefore, while similar, these discussions are not necessarily redundant. No changes made.

8. **P4 L119: typo "dryer"**
   corrected

9. **P8 L245: was it 5% confidence interval or 95%? If 5%, is it reasonable to compare two data sets at a 5% confidence interval?**
   "5% confidence interval" changed to "5% significance level" in both cases, in lines 262 and 266.

10. **P9 L274: typo "HYSPLIT"**
    corrected

11. **Section 3.1.3: the authors should be careful when comparing INP measurements. The size range of INPs, the techniques could be different. For example, Mason et al. (2015) measured INPs using a different technique than PINC and measured INPs up to 10μm.**
    That is correct. In practice, it becomes very difficult to compare [INP] among studies not only due to the size range that was sampled, but also due to other instrumental differences. However, given that the average natural variability over the course of a measurement campaign is so large, the instrumental differences owing to sampled aerosol size cut-offs often contribute negligible error to such comparisons. To acknowledge this, we have added a caveat on lines 317-318, page 11.

12. **Fig.4: how are the size ranges determined? It seems a little bit random. There are some overlapping. Also, does $N_{tot\ >0.5\mu m}$ mean $N_{tot\ 0.5-2.5\mu m}$?**
    We tried to examine as many size channels as possible in an attempt to decipher the effect of particle size on [INP]. The annotations are correct. I.e. $N_{tot\ >0.5\mu m}$ means everything above 0.5 μm. $N_{tot\ 0.5-2.5\mu m}$ means only what is between 0.5 and 2.5μm.

However, $N_{tot\,>0.5\mu m}$ and $N_{tot\,0.5-2.5\mu m}$ are practically identical quantities. No changes made.

**13. P11, L352: a recent paper (Si et al., 2018) correlated the activation fraction with the INP size, which supports your observation here.**
The authors are unsure what is meant by the comment. The discussion in lines 362-375 (page 12) is about the rarity of ambient INP as is highlighted by the presented AF values. We do not discuss the INP size here, only the (very small) number.

**14. Fig. 5 and 6: the capital letters A, B, C are used in the figures, while lower cases a, b, c are used in the text.**
corrected to include capital letters in the text.

**15. Fig. 5A and 6A: how are the back trajectories generated? Does the arrival time still correspond to the mid-point of the INP measurement time? It seems like a new trajectory was generated every 6h.**
In Figs. 5 and 6 the trajectories are generated every six hours during the scenario days in question. This was clarified by modifying the sentence in both sections 3.2.1 and 3.2.2 to read: "The 48-hour trajectory analysis showed that during this time period air masses were arriving from…".

**References:**

Mason, R. H., Si, M., Li, J., Chou, C., Dickie, R., Toom-Sauntry, D., Pöhlker, C., Yakobi-Hancock, J. D., Ladino, L. A., Jones, K., Leaitch, W. R., Schiller, C. L., Abbatt, J. P. D., Huffman, J. A. and Bertram, A. K.: Ice nucleating particles at a coastal marine boundary layer site: correlations with aerosol type and meteorological conditions, Atmos. Chem. Phys., 15(21), 12547–12566, doi:10.5194/acp-15-12547-2015, 2015.

Mason, R. H., Si, M., Chou, C., Irish, V. E., Dickie, R., Elizondo, P., Wong, R., Brintnell, M., Elsasser, M., Lassar, W. M., Pierce, K. M., Leaitch, W. R., MacDonald, A. M., Platt, A., Toom-Sauntry, D., Sarda-Estève, R., Schiller, C. L., Suski, K. J., Hill, T. C. J., Abbatt, J. P. D., Huffman, J. A., DeMott, P. J. and Bertram, A. K.: Size-resolved measurements of ice-nucleating particles at six locations in North America and one in Europe, Atmos. Chem. Phys., 16(3), 1637–1651, doi:10.5194/acp-16-1637-2016, 2016.

Si, M., Irish, V. E., Mason, R. H., Vergara-Temprado, J., Hanna, S. J., Ladino, L. A., Yakobi-Hancock, J. D., Schiller, C. L., Wentzell, J. J. B., Abbatt, J. P. D., Carslaw, K. S., Murray, B. J. and Bertram, A. K.: Ice-nucleating ability of aerosol particles and possible sources at three coastal marine sites, Atmos. Chem. Phys., 18(21), 15669–15685, doi:10.5194/acp-18-15669-2018, 2018.

Welti, A., Müller, K., Fleming, Z. L. and Stratmann, F.: Concentration and variability of ice nuclei in the subtropical maritime boundary layer, Atmos. Chem. Phys., 18(8), 5307– 5320, doi:10.5194/acp-18-5307-2018, 2018.

---

## Referee Report (RR1)

**Re-review of "Condensation/immersion mode ice nucleating particles in a boreal environment"**

Anonymous Reviewer

February 28, 2020

**1  General Comments**

The reviewer would first like to reiterate that the measurements in this work are novel, and should be included in the literature; however, the reviewer still believes that the authors' interpretation of the data, especially the Pearson correlation coefficient analysis in Section 3.2, is incorrect. In their response to this reviewer's main comment, the authors explicitly state "that the Pearson correlation coefficients are not used to chemically speciate the measured INP." Instead, the authors insist that "they are used to infer the predicting capacity of [INP]." This sentiment, however, is not evident from the text in the manuscript. Instead the authors have only added several statements that correlation does not necessarily imply chemical speciation of the INPs; these statements are added as caveats after large sections that imply that correlation does indeed mean causation. With ice nucleation, the sentiment that correlation implies a potential role as an INP is *not* a caveat. It is generally false, and any instance should be scrubbed from the paper. If these sections are allowed to stay in the paper, then other people in the atmospheric chemistry and physics community can cite this paper and incorrectly assume that, for example, 10 to 100-nm biological nanofragments are INPs in boreal forests. This is not supported by the evidence collected in this work. The following are several examples of the authors attempting to use correlation to imply a role as an INP (all line numbers correspond to v4 of the manuscript).

- L26: "On shorter time scales, several particle species correlated well with [INP] implying their potential role as INPs"

- L27/28: "... sub-0.1 $\mu$m particles, most likely nanoscale biological fragments such as ice nucleating macromolecules (INMs), have been found in the INP signal."

- L98: "the investigation of INP physical and chemical properties"

- L332: "In order to probe the identity of the measured INPs ..."

- L411/12: "It seems as though the potential role of BC as INPs active under mixed-phase cloud regime during these particular weather conditions cannot be excluded."

- L417: "Reasons why the signature of BC as an INP shows up only during this time period and not others remain unknown."

- L430: "This supports the general notion that larger, supermicron particles are better INPs"

- L436: "It can, therefore, be said that significant linear correlations of [INP] with supermicron and fluorescent particles, as well as with organics, accompanied by the very high ambient RH likely indicate the importance of these biological particles released by the surrounding forest as INPs"

- L446/7: "However, it is possible that fluorescent particles do contribute to the INP population"

- L465: "What this might mean is that INPs on this day are below 0.1 $\mu$m in diameter, potentially even below 0.01 $\mu$m in diameter."

- L470: "these sub-0.1 $\mu$m particles that are most likely acting as INPs on this particular day."

- L488: "One possible identity of these highly IN-active sub-0.1 $\mu$m particles could be that of nanoscale biological fragments"

- L499/500: "Given the available data, it seems as though the highest measured [INP] during the campaign can be attributed to highly IN-active nanoscale biological fragments originating from surrounding vegetation."

- L527: "On shorter time scales, several particle species correlated well with [INP] implying their potential role as INPs"

- L530-4: "On the day with the highest [INP], sub-0.1 $\mu$m particles, most likely nanoscale biological fragments such as INMs, were found to exhibit a significant correlation with the elevated INP number concentrations. Reasons for why certain particle types act as INPs during certain conditions and not during others and why none of the particle species mentioned above correlate with [INP] across the entire campaign remain unknown."

As the authors note in their–these short term correlations are more likely indicators of specific air masses. Indeed, the authors do a nice job in Section 3.2.1 of stating that the correlation with BC and the back trajectories indicate that this air mass contains INP from biomass burning; however, the extra step in saying that there is a "the signature of BC as an INP" should be avoided. This type of discussion (and avoidance) should be developed in each subsection of section 3.2

**2 General Comments**

- L60: There is a logical fallacy here–black carbon is called a "well known" INP, but their INP activity is then called into question in L69 and L407.

- L63/64: DeMott 2010 only stipulates that n500 is correlated with INP, not that INP are larger. The Mason 2016 reference, however, is correct.

- L70: This mistake was made previously for a reference by Prenni et al., but Petters et al., 2009 only shows that biomass burning can be a source of INP, not BC. In fact, BC are shown *not* to be correlated with INP during the FLAME-2 study.

- L206: Several reviewer comments were centered around the choice of 100 m a.g.l. for the back trajectories. These comments were answered in the author responses to reviewer comments. This should be outlined here–especially the notion that 100 m is the lowest available level in the back trajectory, how well 100 m corresponds to the well-mixed boundary layer, and also the 200 m and 500 m sensitivity studies.

- L388: Interesting that the activated fractions and absolute [INP] are both low here. Does this suggests that biomass burning is not an efficient source of INP (compared to median values at SMEAR II), but could be important regionally?

- L406 and L443: These two lines in this paper correspond to increased CCN activity, however, the CFDC RH is 105%; is there any evidence that these particles do not activate at 5% supersaturation prior to photo-oxidation/sulfate accumulation?

---

## Author Response (AR2)

**Re-review of "Condensation/immersion mode ice nucleating particles in a boreal environment"**

Anonymous Reviewer

February 28, 2020

**1 General Comments**

The reviewer would first like to reiterate that the measurements in this work are novel, and should be included in the literature; however, the reviewer still believes that the authors' interpretation of the data, especially the Pearson correlation coefficient analysis in Section 3.2, is incorrect. In their response to this reviewer's main comment, the authors explicitly state "that the Pearson correlation coefficients are not used to chemically speciate the measured INP." Instead, the authors insist that "they are used to infer the predicting capacity of [INP]." This sentiment, however, is not evident from the text in the manuscript. Instead the authors have only added several statements that correlation does not necessarily imply chemical speciation of the INPs; these statements are added as caveats after large sections that imply that correlation does indeed mean causation. With ice nucleation, the sentiment that correlation implies a potential role as an INP is not a caveat. It is generally false, and any instance should be scrubbed from the paper. If these sections are allowed to stay in the paper, then other people in the atmospheric chemistry and physics community can cite this paper and incorrectly assume that, for example, 10 to 100-nm biological nanofragments are INPs in boreal forests. This is not supported by the evidence collected in this work. The following are several examples of the authors attempting to use correlation to imply a role as an INP (all line numbers correspond to v4 of the manuscript).

- **L26: "On shorter time scales, several particle species correlated well with [INP] implying their potential role as INPs"**
  Authors' response: "implying their potential role as INPs" removed.

- **L27/28: "... sub-0.1 μm particles, most likely nanoscale biological fragments such as ice nucleating macromolecules (INMs), have been found in the INP signal."**
  Authors' response: This sentence has already been corrected in the previous version of the manuscript to say: "…correlated with the INP signal".

- **L98: "the investigation of INP physical and chemical properties"**
  Authors' response: The sentence in question modified to read "The objectives of the campaign included the quantification of [INP] in condensation/immersion freezing modes under mixed-phase cloud conditions, the comparison to previously published data from other locations around the globe and the probing of the predictive capacity of physical and chemical properties on [INP] using correlations."

- **L332: "In order to probe the identity of the measured INPs ..."**
  Authors' response: changed to "In order to infer a predicting capacity of [INP]…"

- **L411/12: "It seems as though the potential role of BC as INPs active under mixed-phase cloud regime during these particular weather conditions cannot be excluded."**

Authors' response: sentence modified to read. "The role of BC as INPs active under mixed-phase cloud conditions needs to be investigated further."

- **L417: "Reasons why the signature of BC as an INP shows up only during this time period and not others remain unknown."**
  Authors' response: Sentence modified to read: "Reasons why BC correlates with the [INP] only during this time period and not others remain unknown."

- **L430: "This supports the general notion that larger, supermicron particles are better INPs"**
  Authors' response: The observed correlation of [INP] with the number of particles 1–2.5 µm in diameter supports the notion that larger, supermicron particles are better INPs. There is nothing wrong with this statement. We observed a correlation, and it corroborates what is already known from published literature.

- **L436: "It can, therefore, be said that significant linear correlations of [INP] with supermicron and fluorescent particles, as well as with organics, accompanied by the very high ambient RH likely indicate the importance of these biological particles released by the surrounding forest as INPs"**
  Authors' response: Sentence modified to read: "It can, therefore, be said that significant linear correlations of [INP] with supermicron and fluorescent particles, as well as with organics, accompanied by the very high ambient RH raise the question whether biological particles released by the surrounding forest could act as INPs."

- **L446/7: "However, it is possible that fluorescent particles do contribute to the INP population"**
  Authors' response: There is nothing wrong with this statement, as it is totally possible that fluorescent particles contribute to the INP population.

- **L465: "What this might mean is that INPs on this day are below 0.1 µm in diameter, potentially even below 0.01 µm in diameter."**
  Authors' response: Sentence deleted.

- **L470: "these sub-0.1 µm particles that are most likely acting as INPs on this particular day."**
  Authors' response: replaced with "these sub-0.1 µm particles that exhibited a strong correlation with [INP] on this particular day"

- **L488: "One possible identity of these highly IN-active sub-0.1 µm particles could be that of nanoscale biological fragments"**
  Authors' response: "highly IN-active" removed.

- **L499/500: "Given the available data, it seems as though the highest measured [INP] during the campaign can be attributed to highly IN-active nanoscale biological fragments originating from surrounding vegetation."**

Authors' response: Sentence modified to read: "Given the available observations, the highest measured [INP] during the campaign correlated best with the concentration of sub-0.1 μm particles. Active INPs in the sub-0.1 μm range have been reported to be from biological particle fragments (Després et al., 2012; Pummer et al., 2012; Pummer et al., 2015)."

- **L527: "On shorter time scales, several particle species correlated well with [INP] implying their potential role as INPs"**
  Authors' response: "implying their potential role as INPs" removed.

- **L530-4: "On the day with the highest [INP], sub-0.1 μm particles, most likely nanoscale biological fragments such as INMs, were found to exhibit a significant correlation with the elevated INP number concentrations. Reasons for why certain particle types act as INPs during certain conditions and not during others and why none of the particle species mentioned above correlate with [INP] across the entire campaign remain unknown."**
  Authors' response: Second sentence modified to read "Reasons for why certain particle types correlated with [INP] during certain conditions and not during others and why none of the particle species mentioned above correlate with [INP] across the entire campaign remain unknown."

**As the authors note in their-these short term correlations are more likely indicators of specific air masses. Indeed, the authors do a nice job in Section 3.2.1 of stating that the correlation with BC and the back trajectories indicate that this air mass contains INP from biomass burning; however, the extra step in saying that there is a "the signature of BC as an INP" should be avoided. This type of discussion (and avoidance) should be developed in each subsection of section 3.2.**

  Authors' response: additional changes have been made in the manuscript to reflect the reviewer's concerns.
- L378-381: sentences modified to read "Therefore, if one is to assume that different particles act as INPs at different times, the predicting capacity of [INP] over the entire duration of the field campaign would not be possible due to averaging. In order to investigate further, the focus is placed on shorter time periods, characterised by particular weather conditions."
- L393-395: sentence modified to read "This situation raises the questions of whether BC is able to act as an INP under our measurement conditions during the examined time period and where this BC may be coming from or if BC is simply an air mass feature."
- L409: "supporting the notion" changed to "supporting the possibility"
- L410-411: sentence modified to read "Additionally, the absence of any other IN-active species may also lead to a clear correlation of BC with [INP], corroborating results presented by Thomson et al. (2018)."
- L498-499: "IN-active" removed.
- L506-508: sentence modified to read "Second, and similar to other special cases examined, it remains unknown why the correlation of [INP] with sub-0.1 μm biological fragments is only visible on this day and not during other periods or during the entire campaign."

**2 General Comments**

- **L60: There is a logical fallacy here-black carbon is called a "well known" INP, but their INP activity is then called into question in L69 and L407.**
  Authors' response: Sentence in L59 modified to read: "The INP activity under different atmospheric conditions has been investigated for several particle species, such as…"

- **L63/64: DeMott 2010 only stipulates that n500 is correlated with INP, not that INP are larger. The Mason 2016 reference, however, is correct.**
  Authors' response: reference to DeMott et al., 2010 removed.

- **L70: This mistake was made previously for a reference by Prenni et al., but Petters et al., 2009 only shows that biomass burning can be a source of INP, not BC. In fact, BC are shown not to be correlated with INP during the FLAME-2 study.**
  Authors' response: reference to Petters et al., 2009 removed.

- **L206: Several reviewer comments were centered around the choice of 100 m a.g.l. for the back trajectories. These comments were answered in the author responses to reviewer comments. This should be outlined here-especially the notion that 100 m is the lowest available level in the back trajectory, how well 100 m corresponds to the well-mixed boundary layer, and also the 200 m and 500 m sensitivity studies.**
  Authors' response: Sentence modified to read: "Trajectories were calculated for the arrival height of 100 m above ground level, which is the arrival height closest to the ground and well-representative of the boundary layer conditions at SMEAR II. Sensitivity tests conducted with trajectory arrival heights of 200 and 500 m did not reveal any differences."

- **L388: Interesting that the activated fractions and absolute [INP] are both low here. Does this suggests that biomass burning is not an efficient source of INP (compared to median values at SMEAR II), but could be important regionally?**
  Authors' response: The low [INP] and AF despite an elevated total particle number concentration may suggest that, while biomass burning may not be an efficient source of INPs, the absence of any other IN-active species at this time may lead to a clear correlation of BC with [INP], as mentioned in L410-411 and supported by a reference to Thomson et al., 2018.

- **L406 and L443: These two lines in this paper correspond to increased CCN activity, however, the CFDC RH is 105%; is there any evidence that these particles do not activate at 5% supersaturation prior to photo-oxidation/sulfate accumulation?**
  Authors' response: No, there isn't any evidence of CCN activity at 243 K in the literature to describe the activation of such particles. In our discussion we suggest that photo-oxidation increases the particle hygroscopicity as shown by an in increase in conventional CCN activity (reported at $T > 273$ K), and how this could lead to an increase in INP activity.

Authors' response references

[revised manuscript text omitted]